# An Efficient Doubly-Robust Test for the Kernel Treatment Effect

**Diego Martinez-Taboada**
Department of Statistics and Data Science
Carnegie Mellon University
Pittsburgh, PA 15213
diegomar@andrew.cmu.edu

**Aaditya Ramdas**
Department of Statistics and Data Science
Machine Learning Department
Carnegie Mellon University
Pittsburgh, PA 15213
aramdas@stat.cmu.edu

**Edward H. Kennedy**
Department of Statistics and Data Science
Carnegie Mellon University
Pittsburgh, PA 15213
edward@stat.cmu.edu

## Abstract

The average treatment effect, which is the difference in expectation of the counterfactuals, is probably the most popular target effect in causal inference with binary treatments. However, treatments may have effects beyond the mean, for instance decreasing or increasing the variance. We propose a new kernel-based test for distributional effects of the treatment. It is, to the best of our knowledge, the first kernel-based, doubly-robust test with provably valid type-I error. Furthermore, our proposed algorithm is computationally efficient, avoiding the use of permutations.

## 1 Introduction

In the context of causal inference, potential outcomes (Rubin, 2005) are widely used to address counterfactual questions (e.g. what would have happened had some intervention been performed?). This framework considers

$$(X, A, Y) \sim \mathbb{P},$$

where $X$ and $Y$ represent the covariates and outcome respectively, and $A \in \{0, 1\}$ is a binary treatment. Furthermore, the following three conditions are assumed:

  i) (Consistency) $Y = AY_1^* + (1 - A)Y_0^*$ (where $Y_1^*, Y_0^*$ are the potential outcomes).

  ii) (No unmeasured confounding) $Y_0^*, Y_1^* \perp\!\!\!\perp A \mid X$.

  iii) (Overlap) For some $\epsilon > 0$, we have $\epsilon < \pi(X) := \mathbb{P}_{A|X}(A = 1|X) < 1 - \epsilon$ almost surely.

Such assumptions allow for identification of causal target parameters. For instance, one may be interested in the average variance-weighted treatment effects (Robins et al., 2008; Li et al., 2011), stochastic intervention effects (Muñoz and Van Der Laan, 2012; Kennedy, 2019), or treatment effect bounds (Richardson et al., 2014; Luedtke et al., 2015). However, most of the literature focuses on estimation and inference of the average treatment effect (Imbens, 2004; Hernan and Robins, 2020), defined as the difference in expectation of the potential outcomes

$$\psi = \mathbb{E}[Y_1^* - Y_0^*].$$

37th Conference on Neural Information Processing Systems (NeurIPS 2023).

Given $(X_i, A_i, Y_i)_{i=1}^N \sim (X, A, Y)$, there are three widespread estimators of $\psi$. First, the plug-in (PI) estimator:

$$\hat{\psi}_{\mathrm{PI}} = \frac{1}{N} \sum_{i=1}^N \left\{ \hat{\theta}_1(X_i) - \hat{\theta}_0(X_i) \right\},$$

where $\hat{\theta}_1(X), \hat{\theta}_0(X)$ estimate $\mathbb{E}[Y_1^*|X], \mathbb{E}[Y_0^*|X]$. Second, the inverse propensity weighting (IPW) estimator:

$$\hat{\psi}_{\mathrm{IPW}} = \frac{1}{N} \sum_{i=1}^N \left\{ \frac{Y_i A_i}{\hat{\pi}(X_i)} - \frac{Y_i(1 - A_i)}{1 - \hat{\pi}(X_i)} \right\},$$

where $\hat{\pi}(X)$ estimates the propensity scores $\mathbb{P}[A = 1|X]$. Third, the so called Augmented Inverse Propensity Weighted (AIPW) estimator:

$$\hat{\psi}_{\mathrm{AIPW}} = \frac{1}{N} \sum_{i=1}^N \left\{ \hat{\theta}_1(X_i) - \hat{\theta}_0(X_i) + \left( \frac{A_i}{\hat{\pi}(X_i)} - \frac{1 - A_i}{1 - \hat{\pi}(X_i)} \right) \left( Y_i - \hat{\theta}_{A_i}(X_i) \right) \right\}.$$

Under certain conditions (e.g., consistent nuisance estimation at $n^{-1/4}$ rates), the asymptotic mean squared error of the AIPW estimator is smaller than that of the IPW and PI estimator, and minimax optimal in a local asymptotic sense (Kennedy, 2022), hence it has become increasingly popular in the last decade. The AIPW estimator is often referred to as the doubly-robust estimator. We highlight that double-robustness is an intriguing property of an estimator that makes use of two models, in which the estimator is consistent even if only one of the two models is well-specified and the other may be misspecified; we refer the reader to Kang and Schafer (2007) for a discussion on doubly-robust procedures.

However, the treatment might have effects beyond the mean, for instance in the variance or skewness of the potential outcome. The average treatment effect will prove insufficient in this case. Consequently, one may be interested in testing whether the treatment has *any* effect in the distribution of the outcome. This question naturally arises in a variety of applications. For instance, one may want to check whether there is any difference between a brand-name drug and its generic counterpart, or understand whether a treatment simply shifts the distribution of the outcome (or, in turn, it also affects higher order moments).

In this work, we revisit the problem of testing the null hypothesis $H_0 : P_{Y_1^*} = P_{Y_0^*}$ against $H_1 : P_{Y_1^*} \neq P_{Y_0^*}$. We propose a distributional treatment effect test based on kernel mean embeddings and the asymptotic behaviour of the AIPW estimator. Our contributions are three-fold:

- Up to our knowledge, we propose the first kernel-based distributional test to allow for doubly-robust estimators with provably valid type-I error.
- The proposed distributional treatment effect test is permutation-free, which makes it computationally efficient.
- We empirically test the power and size of the proposed test, showing the substantial benefits of the doubly robust approach.

## 2   Related work

Distributional treatment effects have been addressed from a variety of points of view. Abadie (2002) was one of the first works to propose to test distributional hypothesis attempting to estimate the counterfactual cumulative distribution function (cdf) of the outcome of the treated and untreated. Chernozhukov et al. (2013) proposed to regress the cdf after splitting the outcome in a grid. Further contributions followed with alike cdf-based approaches (Landmesser, 2016; Díaz, 2017).

Other approaches to the problem include focusing on the probability density function (pdf) instead of the cdf. Robins and Rotnitzky (2001) introduced a doubly robust kernel estimator for the counterfactual density, while Westling and Carone (2020) proposed to conduct density estimation under a monotone density assumption. Kim et al. (2018) and Kennedy et al. (2021) suggested to compute $L^p$ distances between the pdf of the outcome distribution for the different counterfactuals. Conditional distributional treatment effects have also been addressed, with Shen (2019) proposing to estimate the cdf of the counterfactuals for each value to be conditioned on.

On the other hand, kernel methods have recently gained more and more attention in the context of causal inference. Kernel-based two-stage instrumental variable regression was proposed in Singh et al. (2019), while Singh et al. (2020) presented estimators based on kernel ridge regression for nonparametric causal functions such as dose, heterogeneous, and incremental response curves. Furthermore, Singh et al. (2021) conducted mediation analysis and dynamic treatment effect using kernel-based regressors as nuisance functions. Causal inference with treatment measurement error i.e. when the cause is corrupted by error was addressed in Zhu et al. (2022) using kernel mean embeddings to learn the latent characteristic function.

However, kernel mean embeddings for distributional representation were not suggested in the causal inference literature until Muandet et al. (2021) proposed to use an IPW estimator to estimate the average treatment effect on the embedding, which leads to a kernel-based distributional treatment effect test based on this embedding and the MMD. With a very similar motivation, Park et al. (2021) proposed a test for conditional distributional treatment effects based on kernel conditional mean embeddings. In a preprint, Fawkes et al. (2022) extended the work to AIPW estimators, however no theoretical guarantees regarding type-1 error control of the proposed tests were provided.

Finally, the kernel-based tests used in the aforementioned cases involve test statistics that are degenerate U-statistics under the null, hence obtaining theoretical p-values of the statistic is not possible. In turn, Kim and Ramdas (2023) proposed the idea of cross U-statistics, which is based on splitting the data for achieving a normal asymptotic distribution after studentization. Similarly, Shekhar et al. (2022) exploited sample splitting for proposing a permutation-free kernel two sample test.

## 3 Preliminaries

In this section, we introduce the concepts that the proposed test for distributional treatment effects is mainly based on: Maximum Mean Discrepancy (Gretton et al., 2012), Conditional Mean Embeddings (Song et al., 2009), Kernel Treatment Effects (Muandet et al., 2021), dimension-agnostic inference using cross U-statistic (Kim and Ramdas, 2023; Shekhar et al., 2022), and the normal asymptotic behaviour of the doubly robust estimator (Funk et al., 2011).

### 3.1 Maximum Mean Discrepancy (MMD) and Conditional Mean Embeddings

Let $\mathcal{X}$ be a non-empty set and let $\mathcal{H}$ be a Hilbert space of functions $f : \mathcal{X} \to \mathbb{R}$ with inner product $\langle \cdot, \cdot \rangle_{\mathcal{H}}$. A function $k : \mathcal{X} \times \mathcal{X} \to \mathbb{R}$ is called a reproducing kernel of $\mathcal{H}$ if (i) $k(\cdot, x) \in \mathcal{H}$, for all $x \in \mathcal{X}$, (ii) $\langle f, k(\cdot, x) \rangle_{\mathcal{H}} = f(x)$ for all $x \in \mathcal{X}, f \in \mathcal{H}$. If $\mathcal{H}$ has a reproducing kernel, then it is called a Reproducing Kernel Hilbert Space (RKHS).

Building on a reproducing kernel $k$ and a set of probability measures $\mathcal{P}$, the Kernel Mean Embedding (KME) maps distributions to elements in the corresponding Hilbert space as follows:

$$\mu : \mathcal{P} \to \mathcal{H}, \quad \mathbb{P} \to \mu_{\mathbb{P}} := \int k(\cdot, x) d\mathbb{P}(x).$$

If the kernel $k$ is "*characteristic*" (which is the case for frequently used kernels such as the RBF or Matern kernels), then $\mu$ is injective. Conditional mean embeddings (Song et al., 2009) extend the concept of kernel mean embeddings to conditional distributions. Given two RKHS $\mathcal{H}_{k_x}, \mathcal{H}_{k_y}$, the conditional mean embedding operator is a Hilbert-Schimdt operator $\mathcal{C}_{Y|X} : \mathcal{H}_{k_x} \to \mathcal{H}_{k_y}$ satisfying $\mu_{Y|X=x} = \mathcal{C}_{Y|X} k_x(\cdot, x)$, where $\mathcal{C}_{Y|X} := \mathcal{C}_{YX} \mathcal{C}_{XX}^{-1}$, $\mathcal{C}_{YX} := \mathbb{E}_{Y,X}[k_y(\cdot, Y) \otimes k_x(\cdot, X)]$ and $\mathcal{C}_{XX} := \mathbb{E}_{X,X}[k_x(\cdot, X) \otimes k_x(\cdot, X)]$. Given a dataset $\{\mathbf{x}, \mathbf{y}\}$, a sample estimator may be defined as

$$\hat{\mathcal{C}}_{Y|X} = \Phi_{\mathbf{y}}^T (K_{\mathbf{xx}} + \lambda I)^{-1} \Phi_{\mathbf{x}}, \tag{1}$$

where $\Phi_{\mathbf{y}} := [k_y(\cdot, y_1), \ldots, k_y(\cdot, y_n)]^T$, $\Phi_{\mathbf{x}} := [k_x(\cdot, x_1), \ldots, k_x(\cdot, x_n)]^T$, $K_{\mathbf{xx}} := \Phi_{\mathbf{x}} \Phi_{\mathbf{x}}^T$ denotes the Gram matrix, and $\lambda$ is a regularization parameter.

Building on the KME, Gretton et al. (2012) introduced the kernel maximum mean discrepancy (MMD). Given two distributions $\mathbb{P}$ and $\mathbb{Q}$ and a kernel $k$, the MMD is defined as the largest difference in expectations over functions in the unit ball of the respective RKHS:

$$\text{MMD}(\mathbb{P}, \mathbb{Q}) = \sup_{f \in \mathcal{H}} \mathbb{E}_{X \sim \mathbb{P}}[f(X)] - \mathbb{E}_{X' \sim \mathbb{Q}}[f(X')] = ||\mu_{\mathbb{P}} - \mu_{\mathbb{Q}}||_{\mathcal{H}}.$$

It can be shown (Gretton et al., 2012) that $\text{MMD}^2(\mathbb{P}, \mathbb{Q}) = ||\mu_\mathbb{P} - \mu_\mathbb{Q}||_\mathcal{H}^2$. If $k$ is characteristic, then $\text{MMD}(\mathbb{P}, \mathbb{Q}) = 0$ if and only if $\mathbb{P} = \mathbb{Q}$. Given two samples drawn from $\mathbb{P}$ and $\mathbb{Q}$, the MMD between the empirical distributions may be used to test the null hypothesis $H_0 : \mathbb{P} = \mathbb{Q}$ against $H_1 : \mathbb{P} \neq \mathbb{Q}$. However, this statistic is a degenerate two-sample U-statistic under the null, thus one cannot analytically calculate the critical values. Consequently, a permutation-based resampling approach is widely used in practice (Gretton et al., 2012).

### 3.2 Kernel Treatment Effect: A distributional kernel-based treatment effect test

Based on the MMD and the potential outcomes framework, Kernel Treatment Effects (KTE) were introduced in Muandet et al. (2021) for testing distributional treatment effects in experimental settings (i.e. with known propensity scores).

Let $(X_i, A_i, Y_i)_{i=1}^n \sim (X, A, Y)$ such that (i) (consistency) $Y = AY_1^* - (1-A)Y_0^*$, (ii) no unmeasured confounding and overlap assumptions hold. The KTE considers the MMD between $Y_0^*$ and $Y_1^*$ to test $H_0 : \mathbb{P}_{Y_0^*} = \mathbb{P}_{Y_1^*}$ against $H_0 : \mathbb{P}_{Y_0^*} \neq \mathbb{P}_{Y_1^*}$. They define

$$\widehat{\text{KTE}}^2 = ||\hat{\mu}_{Y_1^*} - \hat{\mu}_{Y_0^*}||^2,$$

where

$$\hat{\mu}_{Y_1^*} := \frac{1}{n}\sum_{i=1}^n \frac{A_i k(\cdot, Y_i)}{\pi(X_i)}, \quad \hat{\mu}_{Y_0^*} := \frac{1}{n}\sum_{i=1}^n \frac{(1-A_i)k(\cdot, Y_i)}{1 - \pi(X_i)}.$$

Alternatively, we may also define an unbiased version of it. Again, under the null, these are degenerate two-sample U-statistics. Hence, Muandet et al. (2021) proposed a permutation-based approach for thresholding.

The KTE considers the MMD between the mean embeddings of the counterfactual distributions. This is the cornerstone of the proposed distributional test, as will be exhibited in Section 4, which we extend in a doubly-robust manner to observational settings.

### 3.3 Permutation-free inference using cross U-statistics

The previously mentioned permutation-based approach for obtaining the threshold for the MMD statistic (and hence the KTE statistic) comes with finite-sample validity (Gretton et al., 2012). The number of permutations $B$ used to find the empirical p-values generally varies from 100 to 1000. Consequently, the computational cost of finding a suitable threshold for the statistic is at least $O(Bn^2)$. Such computational cost reduces the applicability of the approach, especially when time or computational resources are limited.

Driven by developing dimension-agnostic inference tools, Kim and Ramdas (2023) presented a permutation-free approach to test null hypotheses of the form $H_0 : \mu = 0$ against $H_1 : \mu \neq 0$, where $\mu$ is the mean embedding of a distribution $\mathbb{P}$, based on the idea of sample splitting. If $X_1, ..., X_{2n} \sim \mathbb{P}$, the usual degenerate V-statistic considers

$$\widehat{\text{MMD}} = ||\hat{\mu}_{2n}||^2,$$

where $\hat{\mu}_{2n} = \frac{1}{2n}\sum_{i=1}^{2n} k(\cdot, X_i)$ (or the similar unbiased version). Kim and Ramdas (2023) proposed to split the data in two and study

$$\widehat{\text{xMMD}} = \langle \hat{\mu}_n^A, \hat{\mu}_n^B \rangle,$$

where $\hat{\mu}_n^A = \frac{1}{n}\sum_{i=1}^n k(\cdot, X_i), \hat{\mu}_n^B = \frac{1}{n}\sum_{j=n+1}^{2n} k(\cdot, X_j)$. Denoting $U_i = \langle k(\cdot, X_i), \hat{\mu}_n^B \rangle$, we have that

$$\widehat{\text{xMMD}} = \frac{1}{n}\sum_{i=1}^n U_i.$$

Under the null and some mild assumptions on the embeddings, Kim and Ramdas (2023, Theorem 4.2) proved that

$$\bar{x}\widehat{\text{MMD}} := \frac{\sqrt{n}\bar{U}}{\hat{S}_u} \xrightarrow{d} N(0, 1),$$

where $\bar{U} = \frac{1}{n}\sum_{i=1}^{n} U_i$, $\hat{S}_u^2 = \frac{1}{n}\sum_{i=1}^{n}(U_i - \bar{U})^2$. Consequently, $\bar{x}\widehat{\text{MMD}}$ is the statistic considered and the null is rejected when $\bar{x}\widehat{\text{MMD}} > z_{1-\alpha}$, where $z_{1-\alpha}$ is the $(1-\alpha)$-quantile of $N(0,1)$. Such test avoids the need for computing the threshold so it reduces the computational cost by a factor of $\frac{1}{B}$, and it is minimax rate optimal in the $L^2$ distance and hence its power cannot be improved beyond a constant factor (Kim and Ramdas, 2023).

The permutation-free nature of cross U-statistic is key in the proposed distributional test. It will allow us to circumvent the need for training regressors and propensity scores repeatedly, while preserving theoretical guarantees.

### 3.4 Empirical mean asymptotic behavior of AIPW

The main property from AIPW estimators that will be exploited in our proposed distributional treatment effect is the asymptotic empirical mean behaviour of the estimator. We present sufficient conditions in the next theorem.

**Theorem 3.1.** *Let* $f(x,a,y) = \{\frac{a}{\pi(x)} - \frac{1-a}{1-\pi(x)}\}\{y - \theta_a(x)\} + \theta_1(x) - \theta_0(x)$, *so that* $\psi = \mathbb{E}\{f(X,A,Y)\}$ *is the average treatment effect. Suppose that*

- $\hat{f}$ *is constructed from an independent sample or* $f$ *and* $\hat{f}$ *are contained in a Donsker class.*

- $||\hat{f} - f|| = o_{\mathbb{P}}(1)$.

*Suppose also that (by clipping)* $\mathbb{P}(\hat{\pi} \in [\epsilon, 1-\epsilon]) = 1$. *If* $||\hat{\pi} - \pi|| \sum_a ||\hat{\theta}_a - \theta_a|| = o_{\mathbb{P}}(\frac{1}{\sqrt{n}})$, *then it follows that*

$$\hat{\psi}_{AIPW} - \psi = (\mathbb{P}_n - \mathbb{P})f(X,A,Y) + o_{\mathbb{P}}(\frac{1}{\sqrt{n}}),$$

*so it is root-n consistent and and asymptotically normal.*

Note that the IPW estimator can be seen as an AIPW with $\hat{\theta}_0(X) = \hat{\theta}_1(X) = 0$ almost surely. The IPW estimator is also asymptotically normal if $||\hat{\pi} - \pi|| = o_{\mathbb{P}}(\frac{1}{\sqrt{n}})$. In experimental settings $||\hat{\pi} - \pi|| = 0$, hence the root-n rate is achieved. Under certain conditions (e.g., consistent nuisance estimation at $n^{-1/4}$ rates), the asymptotic variance of the AIPW estimator is minimized for $\hat{\theta}_1 = \theta_1$, $\hat{\theta}_0 = \theta_0$, thus the IPW estimator is generally dominated by the AIPW if $\hat{\theta}_1, \hat{\theta}_0$ are consistent.

The idea exhibited in Theorem 3.1 will allow for using cross U-statistics in estimated mean embeddings, rather than the actual embeddings. Nonetheless, Theorem 3.1 applies to finite-dimensional outcomes $Y$. We state and prove the extension of Theorem 3.1 to Hilbert spaces in Appendix C, which will be needed to prove the main result of this work.

## 4 Main results

We are now ready to introduce the main result of the paper. Let $Z \equiv (X,A,Y) \sim \mathbb{P}$ be such that $Y = AY_1^* + (1-A)Y_0^*$ and that both no unmeasured confounding and overlap assumptions hold. We denote the space of observations by $\mathcal{Z} = \mathcal{X} \times \mathcal{A} \times \mathcal{Y}$. We are given $Z_i \equiv (X_i, A_i, Y_i)_{i=1}^{2n} \sim (X,A,Y)$ and we wish to test $H_0 : P_{Y_1^*} = P_{Y_0^*}$ against $H_1 : P_{Y_1^*} \neq P_{Y_0^*}$. Given characteristic kernel $k$ i.e. $k(y, \tilde{y}) = \langle k(\cdot, y), k(\cdot, \tilde{y}) \rangle$ with induced RKHS $\mathcal{H}_k$, we equivalently test $H_0 : \mathbb{E}[k(\cdot, Y_1^*) - k(\cdot, Y_0^*)] = 0$.

Under consistency, no unmeasured confounding, and overlap, we have

$$\mathbb{E}[k(\cdot, Y_1^*) - k(\cdot, Y_0^*)] = \mathbb{E}[\phi(Z)],$$

where

$$\phi(z) = \{\frac{a}{\pi(x)} - \frac{1-a}{1-\pi(x)}\}\{k(\cdot, y) - \beta_a(x)\} + \beta_1(x) - \beta_0(x),$$

$$\pi(x) = \mathbb{E}[A \mid X = x], \quad \beta_a(x) = \mathbb{E}[k(\cdot, Y) \mid A = a, X = x].$$

Note the change in notation, from $\theta_a$ to $\beta_a$, to emphasize that such regression functions are now $\mathcal{H}_k$-valued. Thus, we can equivalently test for $H_0 : \mathbb{E}[\phi(Z)] = 0$. With this goal in mind, we denote $\mathcal{D}_1 = (X_i, A_i, Y_i)_{i=1}^n, \mathcal{D}_2 = (X_j, A_j, Y_j)_{j=n+1}^{2n}$ and define

$$T_h^\dagger := \frac{\sqrt{n}\bar{f}_h^\dagger}{S_h^\dagger} \tag{2}$$

where

$$f_h^\dagger(Z_i) = \frac{1}{n} \sum_{j=n+1}^{2n} \langle \hat{\phi}^{(1)}(Z_i), \hat{\phi}^{(2)}(Z_j) \rangle, \ i \in [n].$$

Above, $\hat{\phi}^{(r)}(z)$ is the plug-in estimate of $\phi(z)$ for $r \in \{1, 2\}$ using $\hat{\pi}^{(r)}$ and $\hat{\beta}_a^{(r)}$, which approximate $\pi$ and $\beta_a$ respectively. Further, $\bar{f}_h^\dagger$ and $S_h^\dagger$ denote the empirical mean and standard error of $f_h^\dagger$:

$$\bar{f}_h^\dagger = \frac{1}{n} \sum_{i=1}^n f_h^\dagger(Z_i), \quad S_h^\dagger = \sqrt{\frac{1}{n} \sum_{i=1}^n (f_h^\dagger(Z_i) - \bar{f}_h^\dagger)^2}.$$

The next theorem, which is the main result of the paper, establishes sufficient conditions for $T_h^\dagger$ to present Gaussian asymptotic behavior. While the main idea relies on combining cross U-statistics and the asymptotic empirical mean-like behaviour of AIPW estimators, we highlight a number of technical challenges underpinning this result. The proof combines the central idea presented in Kim and Ramdas (2023) with a variety of techniques including causal inference results, functional data analysis, and kernel method concepts. Furthermore, additional work is needed to extend Theorem 3.1 to $\mathcal{H}_k$-valued outcomes. Donsker classes are only defined for finite dimensional outcomes; in the $\mathcal{H}_k$-valued scenario, we ought to refer to asymptotically equicontinuous empirical processes (Park and Muandet, 2023) and Glivenko-Cantelli classes. We refer the reader to Appendix C for a presentation of such concepts, clarification of the norms used, and the proof of the theorem.

**Theorem 4.1.** *Let $k$ be a kernel that induces a separable RKHS and $\mathbb{P}_{Y_0^*}, \mathbb{P}_{Y_1^*}$ be two distributions. Suppose that (i) $\mathbb{E}[\phi(Z)] = 0$, (ii) $\mathbb{E}\left[\langle \phi(Z_1), \phi(Z_2) \rangle^2\right] > 0$, (iii) $\mathbb{E}\left[\|\phi(Z)\|_{\mathcal{H}}^4\right]$ is finite. For $r \in \{1, 2\}$, suppose that (iv) $\hat{\phi}^{(r)}$ is constructed independently from $\mathcal{D}_r$ or (v) the empirical process of $\hat{\phi}^{(r)}$ is asymptotically equicontinuous at $\phi$ and $\|\hat{\phi}^{(r)}\|_{\mathcal{H}}^2$ belongs to a Glivenko-Cantelli class. If it also holds that (vi) $\|\hat{\phi}^{(r)} - \phi\| = o_{\mathbb{P}}(1)$, (vii) $\mathbb{P}(\hat{\pi}^{(r)} \in [\epsilon, 1 - \epsilon]) = 1$, and*

$$\|\hat{\pi}^{(r)} - \pi\| \sum_a \|\hat{\beta}_a^{(r)} - \beta_a\| = o_{\mathbb{P}}(\frac{1}{\sqrt{n}}) \tag{3}$$

*for $r \in \{1, 2\}$, then it follows that*

$$T_h^\dagger \xrightarrow{d} N(0, 1).$$

We would like to highlight the mildness of the assumptions of Theorem 4.1. The separability of the RKHS is achieved for any continuous kernel on separable $\mathcal{Y}$ (Hein and Bousquet, 2004). Assumption (i) is always attained under the null hypothesis (it is precisely the null hypothesis). Assumption (ii) prevents $\phi(Z)$ from being constant. In such degenerate case, $S_h^\dagger = 0$ thus $T_h^\dagger$ is not even well-defined. Assumption (iii) is the more restrictive out of the first three assumptions, inherited from the use of Lyapunov's CLT in the proof. However, we note that this condition is immediately satisfied under frequently used kernels. For instance, under bounded kernels (for example the common Gaussian and Laplace kernels) such that $\|k(\cdot, Y)\|_{\mathcal{H}} \leq M$, we have that

$$\|\beta_a(x)\|_{\mathcal{H}} = \|\mathbb{E}[k(\cdot, Y)|A = a, X = x]\|_{\mathcal{H}} \leq \mathbb{E}[\|k(\cdot, Y)\|_{\mathcal{H}}|A = a, X = x] \leq M,$$

hence

$$\|\phi(z)\|_{\mathcal{H}} \leq \epsilon^{-1}\|k(\cdot, Y)\|_{\mathcal{H}} + \epsilon^{-1}\max\left(\|\beta_1(x)\|_{\mathcal{H}}, \|\beta_0(x)\|_{\mathcal{H}}\right) + \|\beta_1(x)\|_{\mathcal{H}} + \|\beta_0(x)\|_{\mathcal{H}},$$

which is upper bounded by $2(\epsilon^{-1} + 1)M$. Consequently, $\mathbb{E}\left[\|\phi(Z)\|_{\mathcal{H}}^4\right]^2 \leq \left[2(\epsilon^{-1} + 1)M\right]^8$.

Furthermore, conditions (iv), (vi), (vii) and (3) deal with the proper behaviour of the AIPW estimator; they are standard in the causal inference scenario. Condition (iv) is equivalent to two-fold cross-fitting i.e., training $\hat{\phi}^{(r)}$ on only half of the data and evaluating such an estimator on the remaining half. Condition (v) replaces the Donsker class condition from the finite dimensional setting.

We emphasize the importance of double-robustness in the test; normality of the statistic is achieved due to the $o_{\mathbb{P}}(1/\sqrt{n})$ rate, which is possible in view of the doubly robust nature of the estimators. We also highlight the fact that IPW estimators of the form $\hat{\phi}^{(r)}(z) = \{\frac{a}{\hat{\pi}(x)} - \frac{1-a}{1-\hat{\pi}(x)}\}k(\cdot, y)$ can be embedded in the framework considering $\beta_0, \beta_1 = 0$. In fact, the doubly robust kernel mean embedding estimator may be viewed as an augmented version of the KTE (which is a kernelized IPW) using regression approaches to kernel mean embeddings (Singh et al., 2020), just as AIPW augments IPW with regression approaches. Furthermore, (3) is always attained when the propensity scores $\pi$ are known (i.e. experimental setting), given that $\|\hat{\pi}^{(r)} - \pi\| = 0$.

Based on the normal asymptotic behaviour of $T_h^\dagger$, we propose to test the null hypothesis $H_0 : P_{Y_1^*} = P_{Y_0^*}$ given the p-value $p = 1 - \Phi(T_h^\dagger)$, where $\Phi$ is the cdf of a standard normal. For an $\alpha$-level test, the test rejects the null if $p \leq \alpha$. We consider a one-sided test, rather than studying the two-sided p-value $1 - \Phi(|T_h^\dagger|)$, given that positive values of $T_h^\dagger$ are expected for $\mathbb{E}[\phi(Z)] \neq 0$. The next algorithm illustrates the full procedure of the test, which we call AIPW-xKTE (Augmented Inverse Propensity Weighted cross Kernel Treatment Effect).

---

**Algorithm 1** AIPW-xKTE

1: **input** Data $\mathcal{D} = (X_i, A_i, Y_i)_{i=1}^{2n}$.
2: **output** The p-value of the test.
3: Choose kernel $k$ and estimators $\hat{\beta}_a^{(r)}, \hat{\pi}^{(r)}$ for $r \in \{1, 2\}$.
4: Split data in two sets $\mathcal{D}_1 = (X_i, A_i, Y_i)_{i=1}^n, \mathcal{D}_2 = (X_i, A_i, Y_i)_{i=n+1}^{2n}$.
5: If $\hat{\beta}_a^{(r)}, \hat{\pi}^{(r)}$ are such that condition (v) from Theorem 4.1 is attained, train them on $\mathcal{D}$. Otherwise, train them on $\mathcal{D}_{1-r}$.
6: Define $\hat{\phi}^{(r)}(z) = \{\frac{a}{\hat{\pi}^{(r)}(x)} - \frac{1-a}{1-\hat{\pi}^{(r)}(x)}\}\{k(\cdot, y) - \hat{\beta}_a^{(r)}(x)\} + \hat{\beta}_1^{(r)}(x) - \hat{\beta}_0^{(r)}(x)$.
7: Define $f_h^\dagger(Z_i) = \frac{1}{n}\sum_{j=n+1}^{2n} \langle \hat{\phi}^{(1)}(Z_i), \hat{\phi}^{(2)}(Z_j)\rangle$ for $i = 1, \ldots, n$.
8: Calculate $T_h^\dagger := \frac{\sqrt{n}\bar{f}_h^\dagger}{S_h^\dagger}$, where $\bar{f}_h^\dagger = \frac{1}{n}\sum_{i=1}^n f_h^\dagger(Z_i)$ and $S_h^\dagger = \sqrt{\frac{1}{n}\sum_{i=1}^n (f_h^\dagger(Z_i) - \bar{f}_h^\dagger)^2}$.
9: **return** p-value $p = 1 - \Phi(T_h^\dagger)$.

---

Note that the proposed statistic is, at heart, a two sample test (with a nontrivial causal twist); in contrast to Shekhar et al. (2022), the two samples are not independent and are potentially confounded.

Extensive literature focuses on designing estimator $\hat{\pi}^{(r)}$, logistic regression being the most common choice. At this time, not so many choices exist for estimators $\hat{\beta}_a^{(r)}$, given that it involves a regression task in a Hilbert space. Conditional mean embeddings are the most popular regressor, although other choices exist (Ćevid et al., 2022).

Note that we have motivated the proposed procedure for testing distributional treatment effects with characteristic kernels. However, the actual null hypothesis being tested is $H_0 : \mathbb{E}[k(\cdot, Y_1^*)] = \mathbb{E}[k(\cdot, Y_0^*)]$. If the kernel chosen is not characteristic, the test would continue to be valid for $H_0$, although it would not be valid to test equality between $\mathbb{P}_{Y_0^*}$ and $\mathbb{P}_{Y_1^*}$. For instance, AIPW-xKTE with a linear kernel could be used to test equality in means of counterfactuals.

Furthermore, the proposed test is permutation-free, as the statistic $T_h^\dagger$ ought to be computed only once. This permutation-free nature is crucial, as it avoids the repeated estimation of $\hat{\pi}^{(r)}, \hat{\beta}_a^{(r)}$. For instance, conditional mean embeddings involve the inversion of a matrix, which scales at least at $O(n^\omega)$, with practical values being $\omega = 2.87$ by Strassen's algorithm (Strassen et al., 1969). Calculating the conditional mean embedding for every permutation would imply $O(Bn^\omega)$, where $B$ is the number of permutations. Furthermore, regressors for mean embeddings of different nature might involve a higher complexity, hence avoiding permutations becomes even more important in the approach.

If the actual embedding $\phi$ was known, the power of AIPW-xKTE could not be improved beyond a constant factor (by minimax optimality of cross U-statistics in $L^2$ distance). Further, every procedure will suffer from the error in estimation of $\phi$. This means that we are potentially incurring in a loss of power by avoiding a permutation-based approach, however such a loss is controlled by a small factor. Nonetheless, this potential loss is inherited from splitting the data in our estimator (only half of the data is used on each side of the inner product). We highlight that sample splitting is needed when using flexible doubly-robust estimators, hence we expect no loss in power compared to other potential doubly-robust approaches in that case.

## 5    Experiments

In this section, we explore the empirical calibration and power of the proposed test AIPW-xKTE. For this, we assume that we observe $(x_i, a_i, y_i)_{i=1}^n \sim (X, A, Y)$ and that (causal inference assumptions) consistency, no unmeasured confounding, and overlap hold. Both synthetic data and real data are evaluated. All the tests are considered at a 0.05 level. For an exhaustive description of the simulations and outcomes, including additional experiments, we direct the reader to Appendix B.

**Synthetic data.** All data (covariates, treatments and responses) are artificially generated. We define four scenarios:

- Scenario I: There is no treatment effect; thus, $\mathbb{P}_{Y_0^*} = \mathbb{P}_{Y_1^*}$.
- Scenario II: There exists a treatment effect that only affects the means of $\mathbb{P}_{Y_0^*}, \mathbb{P}_{Y_1^*}$.
- Scenario III and Scenario IV: There exists a treatment effect that does not affect the means but only affects the higher moments of $\mathbb{P}_{Y_0^*}$ and $\mathbb{P}_{Y_1^*}$, differently for each scenario.

For all four scenarios, we consider the usual observational study setting, where the propensity scores $\pi(X)$ are treated as unknown and hence they must be estimated. We define the proposed AIPW-xKTE test with the mean embedding regressions fitted as conditional mean embeddings and the propensity scores estimated by logistic regression.

We first study the empirical calibration of AIPW-xKTE and the Gaussian behaviour of $T_h^\dagger$ under the null. Figure 1 exhibits the performance of AIPW-xKTE in Scenario I. Both a standard normal behaviour and proper calibration are empirically attained in the simulations.

Due to the fact that the KTE (Muandet et al., 2021) may not be used in the observational setting, where the propensity scores are not known, there is no natural benchmark for the proposed test. In particular, we were unable to control the type-1 error of the test presented in Fawkes et al. (2022), and hence omitted from our simulations. Consequently, we compare the power of the proposed AIPW-xKTE and IPW-xKTE with respect to three methods that are widely used while conducting inference on the average treatment effect: Causal Forests (Wager and Athey, 2018), Bayesian Additive Regression Trees (BART) (Hahn et al., 2020), and a linear regression based AIPW estimator (Baseline-AIPW).

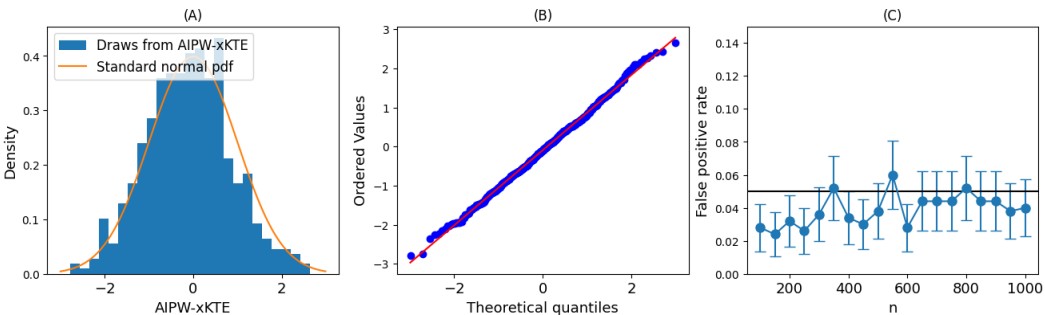

Figure 1: Illustration of 500 simulations of the AIPW-xKTE under the null: (A) Histogram of AIPW-xKTE alongside the pdf of a standard normal for $n = 500$, (B) Normal Q-Q plot of AIPW-xKTE for $n = 500$, (C) Empirical size of AIPW-xKTE against different sample sizes. The figures show the Gaussian behaviour of the statistic under the null, which leads to a well calibrated test.

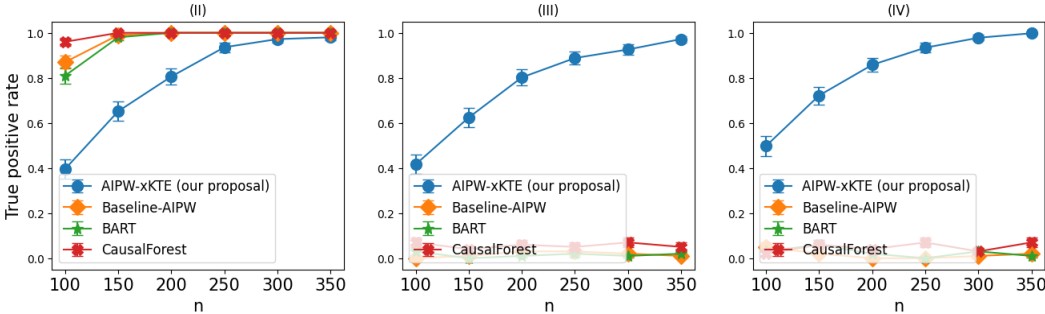

Figure 2: True positive rates of 500 simulations of the tests in Scenarios II, III, and IV. AIPW-xKTE shows notable true positive rates in every scenario, unlike competitors.

Figure 2 exhibits the performance of such tests in Scenario II, Scenario III, and Scenario IV. The three methods dominate AIPW-xKTE in Scenario II, where there exists a mean shift in counterfactuals. However, and as expected, such methods show no power if the distributions differ but have equal means. In contrast, AIPW-xKTE detects distributional changes beyond the mean, exhibiting power in all scenarios.

*Remark:* While this work focuses on the observational setting, where double robustness is crucial, we highlight that the proposed AIPW-xKTE test may also be used in experiments (where propensity scores are known) for computational gains. The proposed test avoids permutations, which makes it more computationally efficient than the KTE (Muandet et al., 2021). We refer the reader to Appendix A for a comparison between the proposed AIPW-xKTE and the KTE.

**Real data.** We use data obtained from the Infant Health and Development Program (IHDP) and compiled by Hill (2011), in which the covariates come from a randomized experiment studying the effects of specialist home visits on cognitive test scores. This data has seen extensive use in causal inference (Johansson et al., 2016; Louizos et al., 2017; Shalit et al., 2017). We work with 18 variables of the covariate set and unknown propensity scores. We highlight that the propensity score model is likely misspecified in this real life scenario.

We consider six scenarios with the IHDP data. For Scenarios I, II, III and IV, we generate the response variables similarly to the previous experiments. In Scenario V, we take the IQ test (Stanford Binet) score measured at the end of the intervention (age 3) as our response variable. For Scenario VI, we calculate the average treatment effect in Scenario V using Causal Forests (obtaining 0.003 i.e. a positive shift) and subtract it from the IQ test score of those who are treated, thus obtaining a distribution of the IQ test scores with zero average treatment effect.

For each of the scenarios, we consider the tests AIPW-xKTE, Causal Forests, BART and Baseline-AIPW on 500 bootstrapped subsets to estimate the rejection rates. All of them showed expected levels of rejection under the null i.e. Scenario I. The results for the remaining scenarios can be found in Table 1. We note that the performance of the tests on Scenarios I, II, III, and IV is similar to the analogous scenarios on synthetic data. While AIPW-xKTE exhibits a loss in power with respect to

Table 1: True positive rates ($\pm$ std) for the different scenarios and tests using the IHDP data. While AIPW-xKTE shows less power in Scenario II, it outperforms its competitors in Scenarios III and IV.

| Test | Scenario | | | | |
|---|---|---|---|---|---|
| | II | III | IV | V | VI |
| AIPW-xKTE | $0.44 \pm 0.05$ | $0.34 \pm 0.05$ | $0.53 \pm 0.05$ | $0.99 \pm 0.01$ | $0.03 \pm 0.02$ |
| Baseline-AIPW | $0.95 \pm 0.02$ | $0.00 \pm 0.00$ | $0.00 \pm 0.00$ | $1.00 \pm 0.00$ | $0.00 \pm 0.00$ |
| BART | $0.77 \pm 0.04$ | $0.15 \pm 0.04$ | $0.24 \pm 0.04$ | $0.10 \pm 0.03$ | $0.04 \pm 0.02$ |
| CausalForest | $1.00 \pm 0.00$ | $0.04 \pm 0.02$ | $0.05 \pm 0.02$ | $1.00 \pm 0.00$ | $0.00 \pm 0.00$ |

the other tests when the average treatment effect is non zero, it detects distributional changes beyond the mean.

Besides BART, all of the tests reject the null in almost every simulation of Scenario V. As expected, Causal Forest, BART and Baseline-AIPW barely reject the null in Scenario VI, which considers the data of Scenario V with zero average treatment effect. Interestingly, the proposed distributional test has a rejection rate below 0.05, which supports the fact that specialist home visits have no effect on the distribution of cognitive test scores beyond an increase in the mean.

## 6   Conclusion and future work

We have developed a computationally efficient kernel-based test for distributional treatment effects. It is, to our knowledge, the first kernel-based test for distributional treatment effects that allows for a doubly-robust approach with provably valid type-I error. Furthermore, it does not suffer from the computational costs inherent to permutation-based tests. The proposed test empirically proves valid in the observational setting, where its predecessor KTE may not be used: the test is well calibrated and shows power in a variety of scenarios. Procedures designed to test for average treatment effects only outperform the proposed test if there is a mean shift between counterfactuals.

There are several possible avenues for future work. We highlighted that our procedure holds if consistency, no unmeasured confounding, and overlap hold. However, this may often not be the case in observational studies. Generalizing the work to other causal inference frameworks, for example by considering instrumental variables, would be of interest. Exploring the extension of this work to test for conditional treatment effects could also be a natural direction to follow. Lastly, we expect state-of-the-art regressors for kernel mean embeddings, such as distributional random forests (Ćevid et al., 2022), to find outstanding use in our estimator. We envisage that this may motivate the development of flexible mean embedding estimators designed for complex data, such as image or text.

## Acknowledgements

The project that gave rise to these results received the support of a fellowship from 'la Caixa' Foundation (ID 100010434). The fellowship code is LCF/BQ/EU22/11930075.

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

# A Comparison between KTE and AIPW-xKTE in the experimental setting

In Section 5, we exhibited the performance of the proposed AIPW-xKTE test in the observational setting. We emphasize that the proposed AIPW-xKTE test is designed to target the observational setting, where the KTE may not be used given that the propensity scores are unknown. In such a setting, the double-robustness of the proposed test is fully exploited. However, the proposed test may also be used in the experimental setting for its computational efficiency: the AIPW-xKTE test avoids permutations, which makes it more computationally efficient than the KTE.

Consequently, we explore in this appendix the performance of the AIPW-xKTE in the experimental setting. We consider a usual experimental design where $\frac{n}{2}$ units are treated and $\frac{n}{2}$ units are not treated so that $\pi(X) = \frac{1}{2}$. We compare the power of the proposed AIPW-xKTE with respect to the KTE test. Further, we include an IPW version of the proposed test (by taking $\beta_0 = \beta_1 = 0$, see comments in Section 4) in order to elucidate the interest in considering the doubly robust version. We refer the reader to Appendix B for an exhaustive description of the experiments.

Figure 4 exhibits the performance of the tests in Scenario II, Scenario III, and Scenario IV. We note that IPW-xKTE has less power than the KTE, which is its analogous permutation-based version. This is due to the controlled loss in power of the cross U-statistic approach following the comments in Section 4. However, we see that the AIPW-xKTE competes with the KTE, despite showing slightly less power. The loss in power inherited by the permutation-free approach is compensated by the gain in power due to using an AIPW estimator. Furthermore, AIPW-xKTE shows a drastic improvement in computational costs with respect to KTE, as exhibited in Table 2. Lastly, we see that AIPW-xKTE clearly outperforms IPW-xKTE, which illustrates the significant benefits of the doubly robust approach.

Table 2: Average times (in milliseconds) for different tests and sample sizes. AIPW-xKTE is almost 20 times faster than KTE for $n = 350$ with only 100 permutations used.

| Test | $n$ | | |
|---|---|---|---|
| | 150 | 250 | 350 |
| AIPW-xKTE | 1.770 | 2.994 | 4.940 |
| IPW-xKTE | 1.496 | 2.297 | 2.331 |
| KTE | 16.903 | 44.483 | 89.875 |

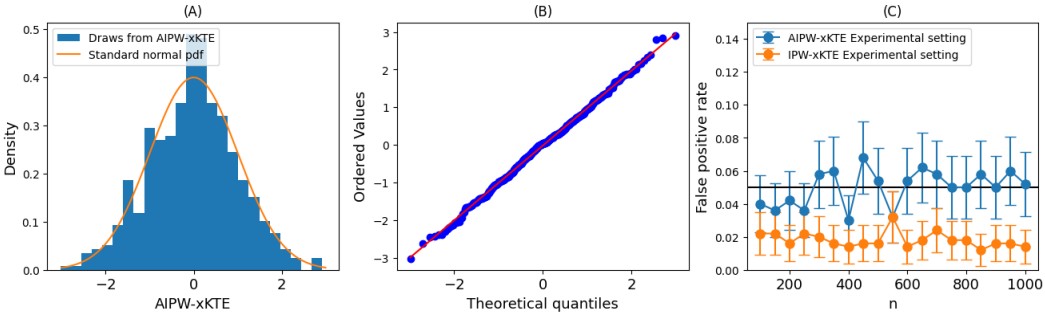

Figure 3: Illustration of 500 simulations of the AIPW-xKTE under the null in the experimental setting: (A) Histogram of AIPW-xKTE alongside the pdf of a standard normal for $n = 500$, (B) Normal Q-Q plot of AIPW-xKTE for $n = 500$, (C) Empirical size of AIPW-xKTE and IPW-xKTE against different sample sizes. The figures show the Gaussian behaviour of the statistic under the null, which leads to a well calibrated test.

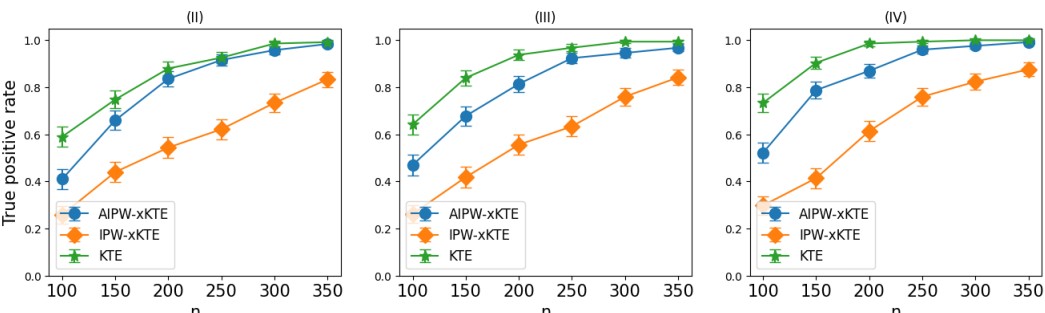

Figure 4: True positive rates of 500 simulations of the tests in Scenarios II, III, and IV in the experimental setting. AIPW-xKTE demonstrates comparable power to KTE, and improves appreciably over IPW-xKTE.

# B Experiments

We present in this appendix a comprehensive display of the simulations conducted. Subsection B.1 includes an exhaustive description of the experiments introduced in Section 5 and Appendix A. Additional simulations and results may be found in Subsection B.2.

## B.1 Exhaustive description of experiments

We assume that we observe $(x_i, a_i, y_i)_{i=1}^n \sim (X, A, Y)$ and that (causal inference assumptions) consistency, no unmeasured confounding, and overlap hold. All the tests are considered at a 0.05 level. Both synthetic data and real data are evaluated.

**Synthetic data.** All the data (covariates, treatments and responses) is artificially generated. We define

$$Y_0^* = \beta^T X + \epsilon_0, Y_1^* = \beta^T X + b + \epsilon_1, \tag{4}$$

such that $\epsilon_0, \epsilon_1 \sim N(0, 0.5)$ are independent noises, $X \sim N(0, I_5)$ and $\beta = [0.1, 0.2, 0.3, 0.4, 0.5]^T$. We set b = 0 and b = 2 for Scenario I and Scenario II respectively. For Scenario III, we set $b = 2Z - 1$, where $Z$ is an independent Bernoulli random variable $Z \sim \text{Bernoulli}(0.5)$. In Scenario IV, $b \sim \text{Uniform}(-2, 2)$.

In the experimental setting, we consider a usual experimental design where $\frac{n}{2}$ units are treated and $\frac{n}{2}$ units are not treated such that $\pi(X) = \frac{1}{2}$ almost surely. In the observational setting, we define $\pi(X) = s(\alpha^T X + \alpha_0)$, where $s(z) = 1/(1 + exp(-z))$ (sigmoid function), $\alpha_0 = 0.05$ and $\alpha = [0.05, 0.04, 0.03, 0.02, 0.01]^T$. In such setting, we estimate $\hat{\pi}$ using an L2 regularized logistic regression with the regularization term set to 1e-6. For AIPW-xKTE, the mean embedding regressions are fitted as conditional mean embeddings.

In the experimental setting, we consider the tests IPW-xKTE, AIPW-xKTE, and KTE (Muandet et al., 2021). In the observational setting, we consider IPW-xKTE, AIPW-xKTE and permutation-based tests based on Causal Forests (Wager and Athey, 2018), Bayesian Additive Regression Trees (BART) (Hahn et al., 2020), and a linear regression based AIPW estimator (Baseline-AIPW). For the latter three, we recalculate the respective statistics for every permutation and reject the null if the original statistic is above the 0.95 empirical quantile. For all scenarios and settings, we consider 500

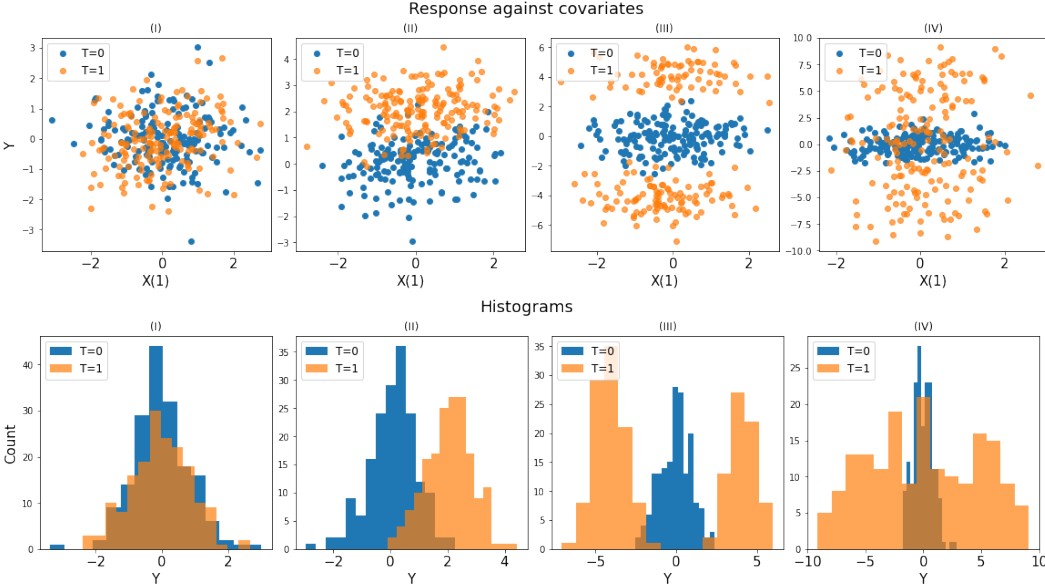

Figure 5: Illustration of one simulation of the simulated data in the experimental setting with $n = 350$ for Scenarios I, II, III, and IV. We display scatter plots of the outcome against the first covariate and histograms of the outcomes, both grouped by treatment.

simulations for each $n \in \{100, 150, 200, 250, 300, 350\}$. We exhibit an illustration of the synthetic data from one the simulations carried in Figure 5.

**Real data.** We use data obtained from the Infant Health and Development Program (IHDP) and compiled by Hill (2011), in which the covariates come from a randomized experiment studying the effects of specialist home visits on cognitive test scores. The propensity scores are unknown and hence they must be estimated. We work with the following 18 variables of the covariate set: ['bw','momage','nnhealth','birth.o','parity','moreprem','cigs','alcohol','ppvt.imp', 'bwg','female','mlt.birt','b.marry','livwho','language','whenpren','drugs','othstudy'], where the first nine are continuous and the last nine are discrete; we refer to Hill (2011) for a detailed presentation of the data set. We eliminate the rows in which there is a NaN for any of the variables considered, finishing with 908 observations, out of which 347 were treated. Further, we standarize the data such that every continuous variable has mean 0 and variance 1. We illustrate the data obtained after preprocessing in Figure 6.

We consider five scenarios with the IHDP data. For Scenarios I, II, III and IV, we generate the response variables similarly to the previous experiments:

$$Y_0^* = \beta^T X + \epsilon_0, Y_1^* = \beta^T X + b + \epsilon_1, \tag{5}$$

where $\epsilon_0, \epsilon_1 \sim N(0, 0.5)$ are independent noises, and $\beta = [1, \dots, 1]^T$. We set b = 0 and b = 1 for Scenario I and Scenario II respectively. For scenario III, we set $b = 2(2Z - 1)$, where $Z$ is an independent Bernoulli random variable $Z \sim \text{Bernoulli}(0.5)$. In scenario IV, $b \sim \text{Uniform}(-4, 4)$. In Scenario V, we take the IQ test (Stanford Binet) score, variable 'iqsb.36' of the data set, measured at the end of the intervention (age 3) as our response variable. For Scenario VI, we calculate the average treatment effect in Scenario V using Causal Forests (obtaining 0.0035 i.e. a positive shift) and subtract it to the IQ test score of those who are treated, thus obtaining a distribution of the IQ test scores with zero average treatment effect. We illustrate the data of Scenarios V and VI in Figure 6. For each of the scenarios, we consider the tests AIPW-xKTE, IPW-xKTE, Causal Forests, BART and Baseline-AIPW on 500 bootstrapped subsets to estimate the rejection rates. For AIPW-xKTE and IPW-xKTE, we estimate $\hat{\pi}$ using a regularized logistic regression with the regularization term set to 1e-6. For AIPW-xKTE, the mean embedding regressions are fitted as conditional mean embeddings.

We sample split for estimating the conditional mean embeddings, but for the propensity scores, we use the whole training data. While we stated Theorem 4.1 imposing condition (iv) *or* condition (v) on $\hat{\phi}^{(r)}$ (common practice in causal inference for ease of presentation, where one usually imposes

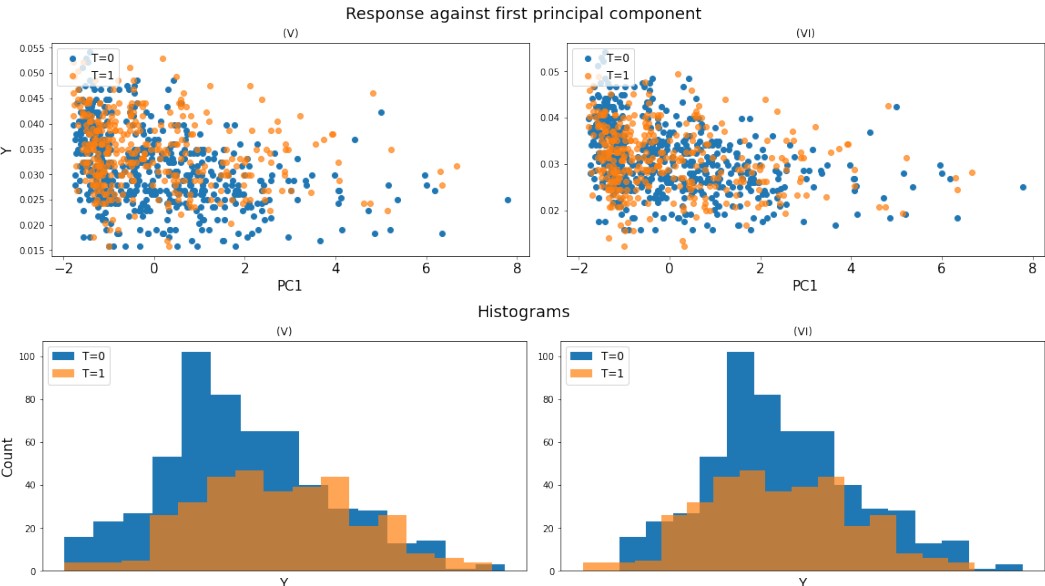

Figure 6: Illustration of the IHDP data for Scenarios V and VI. We display scatter plots of the outcome against the first principal component and histograms of the outcomes, both grouped by treatment.

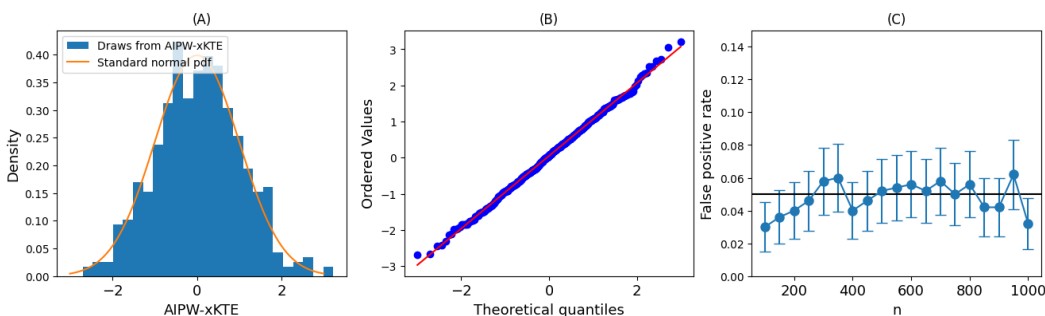

Figure 7: Illustration of 500 simulations of the AIPW-xKTE with non-linear effects under the null: (A) Histogram of AIPW-xKTE alongside the pdf of a standard normal for $n = 500$, (B) Normal Q-Q plot of AIPW-xKTE for $n = 500$, (C) Empirical size of AIPW-xKTE against different sample sizes. The figures show the same Gaussian behaviour of the statistic under the null as in the linear setting.

sample splitting *or* a Donsker condition), Theorem 4.1 holds as long as sample splitting is conducted for those estimators $\hat{\pi}^{(r)}, \hat{\beta}_a^{(r)}$ which may overfit (analogous case in the standard doubly robust estimator). In our case, the propensity is modeled by L2 regularized logistic regression, which is simple and cannot overfit when the dimension is fixed and $n \to \infty$.

All the kernels considered on $\mathcal{Y}$ for AIPW-xKTE, IPW-xKTE, and KTE are RBF i.e. $k_y(y_1, y_2) = exp(\nu_y |y_1 - y_2|^2)$, with scale parameter $\nu_y$ chosen by the median heuristic. For AIPW-xKTE, the mean embedding regressions are fitted as conditional mean embeddings, and we conduct sample splitting in order to train them. The kernel considered for such conditional mean embeddings on $\mathcal{X}$ is also RBF $k_x(x_1, x_2) = exp(\nu_x |x_1 - x_2|^2)$ with scale parameter $\nu_x$ chosen by the median heuristic as well. The regularization parameter $\lambda$ was taken to be equal to $\nu_x$.

## B.2 Further experiments

In this subsection, we investigate the performance of the tests in non-linear settings. We extend the simulations with synthetic data presented in Subsection B.1, defining the potential outcomes

$$Y_0^* = \cos(\beta^T X) + \epsilon_0, \quad Y_1^* = \cos(\beta^T X) + b + \epsilon_1, \tag{6}$$

where $\beta$, $b$, $\epsilon_0$ and $\epsilon_1$ are defined as described in Subsection B.1. The different scenarios and the remaining parameters are also defined as in Subsection B.1.

Figure 7 displays the behaviour of AIPW-xKTE in Scenario I (under the null). Analogously to the linear setting, we note that AIPW-xKTE presents a Gaussian behaviour under the null, which leads to a well-calibrated test. The performance of the tests in Scenarios II, III and IV is displayed in Figure 8. In this non-linear case, AIPW-xKTE proves even more competitive in Scenario II, while retaining power in Scenarios III and IV. In short, the test proves valid (as expected) in the non-linear case, and the comments from Section 5 equally apply.

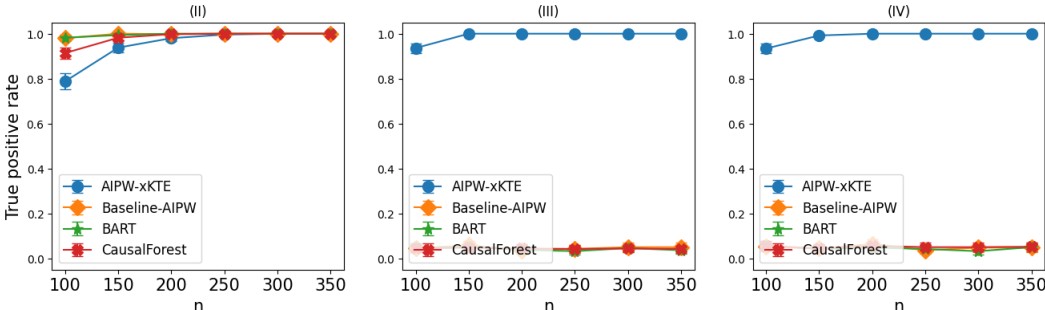

Figure 8: True positive rates of 500 simulations of the tests in Scenarios II, III, and IV with non-linear effects. AIPW-xKTE shows notable true positive rates in every scenario, as in the linear setting.

## C Proofs

We present in this appendix the proof of the main result of the paper, Theorem 4.1. For this purpose, we introduce the notation used in Subsection C.1 followed by the exposition of previously known results in Subsection C.2. We present a sequence of extensions of such results to the vector-valued scenario in Subsection C.3, which will be used to prove the main theorem of the paper in Subsection C.4.

### C.1 Notation

We use standard big-oh and little-oh notation, i.e., $X_n = O_{\mathbb{P}}(r_n)$ means $X_n/r_n$ is bounded in probability and $X_n = o_{\mathbb{P}}(r_n)$ means $X_n/r_n \xrightarrow{p} 0$. We use $\mathbb{P}_n$ for sample averages i.e. $\mathbb{P}_n(f) = \mathbb{P}_n\{f(Z)\} = \frac{1}{n}\sum_i f(Z_i)$. For a potentially random function $\widehat{f}$, we denote $\mathbb{P}(\widehat{f}) = \mathbb{P}\{\widehat{f}(Z)\} = \int \widehat{f}(z)\,d\mathbb{P}(z)$, and $\|\widehat{f}\|^2 = \int \widehat{f}(z)^2\,d\mathbb{P}(z)$ represents the squared $L_2(\mathbb{P})$ norm.

Given a Hilbert space $\mathcal{H}$ and a $\mathcal{H}$-valued $\omega(z) \in \mathcal{H}$, we denote its norm in the Hilbert space by $\|\omega(z)\|_{\mathcal{H}}$. We let $\mathbb{P}_n(\omega) = \mathbb{P}_n\{\omega(Z)\} = \frac{1}{n}\sum_i \omega(Z_i)$ and $\mathbb{P}(\widehat{\omega}) = \mathbb{P}\{\widehat{\omega}(Z)\} = \int \widehat{\omega}(z)\,d\mathbb{P}(z)$ (Bochner integral). Furthermore, $\|\widehat{\omega}\|^2 = \int \|\widehat{\omega}(z)\|_{\mathcal{H}}^2\,d\mathbb{P}(z)$ denotes the squared $L_2(\mathbb{P})$ norm of the $\mathcal{H}$-valued function norm, and $\|\cdot\|_{\mathrm{HS}}$ the Hilbert-Schmidt norm of an operator.

### C.2 Previously known results

We start by introducing a generalization of the formula that relates the mean, variance and second moment of a real-valued random variable to $\mathcal{H}$-valued random variables.

**Lemma C.1** (Hsing and Eubank (2015, Theorem 7.2.2)). *Let $\omega \in \mathcal{H}$ be a random variable such that $\mathbb{E}\|\omega\|_{\mathcal{H}}^2 < \infty$. Then*

$$\mathbb{E}\|\omega - \mathbb{E}[\omega]\|_{\mathcal{H}}^2 = \mathbb{E}\|\omega\|_{\mathcal{H}}^2 - \|\mathbb{E}[\omega]\|_{\mathcal{H}}^2.$$

Furthermore, one may retrieve the expectation of the product of projections of the $\mathcal{H}$-valued random variables through the covariance operator, as exhibited in the following lemma.

**Lemma C.2.** *Let $\omega \in \mathcal{H}$ be a random variable such that $\mathbb{E}[\omega] = 0$ and $\mathbb{E}\|\omega\|_{\mathcal{H}}^2 < \infty$. Then*

$$\mathbb{E}[\langle \omega, f\rangle\langle \omega, g\rangle] = \langle Cf, g\rangle, \quad f, g \in \mathcal{H},$$

*where $C = \mathbb{E}[\omega \otimes \omega]$ is the covariance operator.*

Next, we present the generalization of the central limit theorem to $\mathcal{H}$-valued random variables.

**Theorem C.3** (Bosq (2000, Theorem 2.7)). *Let $(\omega_i, i \geq 1)$ be a sequence of iid random variables in a separable Hilbert space $\mathcal{H}$. If $m = \mathbb{E}[\omega_1]$ and $\mathbb{E}[\|\omega_1\|^2] < \infty$, then*

$$\frac{1}{\sqrt{n}}\sum_{i=1}^n (\omega_i - m) \xrightarrow{d} \xi,$$

*where $\xi$ is a Gaussian random element with mean zero and covariance operator $C = \mathbb{E}[\omega_1 \otimes \omega_1]$ i.e.*

$$\xi \sim N(0, \mathbb{E}[\omega_1 \otimes \omega_1]).$$

Moreover, Gaussian $\mathcal{H}$-valued random variables admit the following expansion in the orthonormal system defined by its covariance operator.

**Lemma C.4** (Horváth and Kokoszka (2012, Equation 2.8)). *A normally distributed function $\xi$ in a separable Hilbert space with a covariance operator $C$ admits the expansion*

$$\xi \stackrel{d}{=} \sum_{i}^{\infty} \sqrt{\lambda_j} N_j v_j,$$

*where $N_j$ are independent standard normal distributions, $(\lambda_j, j \geq 1)$ is the sequence of eigenvalues of the covariance operator $C$, and $(v_j, j \geq 1)$ is a sequence of orthonormal eigenfunctions of the covariance operator $C$.*

We now present a result concerning the asymptotic behaviour of an empirical process given that the considered random functions belong to a Donsker class. For a full discussion on Donsker classes and empirical processes, we refer the reader to Chapter 18 and Chapter 19 of Van der Vaart (2000).

**Lemma C.5** (Van der Vaart (2000, Lemma 19.24)). *Suppose that $\mathcal{F}$ is a $\mathbb{P}$-Donsker class of measurable (real-valued) functions and $\hat{f}_n$ is a sequence of random functions that take their values in $\mathcal{F}$ such that $\|\hat{f}_n - f\|^2 \xrightarrow{P} 0$ for some $f \in L_2(\mathbb{P})$. Then*

$$(\mathbb{P}_n - \mathbb{P})(\widehat{f}_n - f) = o_{\mathbb{P}}\left(\frac{1}{\sqrt{n}}\right).$$

However, the concept of Donsker class is fairly restrictive. In fact, the intuition behind the nice behaviour exhibited by Donsker classes in empirical process is their inability to overfit, due to their constricted flexibility. For instance, commonly used estimators such as random forests do not belong to a Donsker class. In order to circumvent this problem, sample splitting proves very useful. The following lemma exhibits the power of sample splitting regardless of the flexibility of the estimators.

**Lemma C.6** (Kennedy et al. (2020, Lemma 2)). *Let $\widehat{f}(z)$ be a function estimated from a sample $Z^N = (Z_{n+1}, \ldots, Z_N)$, and let $\mathbb{P}_n$ denote the empirical measure over $(Z_1, \ldots, Z_n)$, which is independent of $Z^N$. Then*

$$(\mathbb{P}_n - \mathbb{P})(\widehat{f} - f) = O_{\mathbb{P}}\left(\frac{\|\widehat{f} - f\|}{\sqrt{n}}\right).$$

*Proof.* First note that, conditional on $Z^N$, the term in question has mean zero since

$$\mathbb{E}\left\{(\mathbb{P}_n - \mathbb{P})(\widehat{f} - f) \;\Big|\; Z^N\right\} = \mathbb{E}\left\{\mathbb{P}_n(\widehat{f} - f) \;\Big|\; Z^N\right\} - \mathbb{P}(\widehat{f} - f)$$
$$= \mathbb{E}(\widehat{f} - f \mid Z^N) - \mathbb{P}(\widehat{f} - f)$$
$$= \mathbb{P}(\widehat{f} - f) - \mathbb{P}(\widehat{f} - f)$$
$$= 0.$$

The conditional variance is

$$\text{var}\left\{(\mathbb{P}_n - \mathbb{P})(\widehat{f} - f) \;\Big|\; Z^N\right\} = \text{var}\left\{\mathbb{P}_n(\widehat{f} - f) \;\Big|\; Z^N\right\} = \frac{1}{n}\text{var}(\widehat{f} - f \mid Z^N) \leq \|\widehat{f} - f\|^2/n.$$

Therefore by iterated expectation and Chebyshev's inequality we have

$$\mathbb{P}\left\{\frac{|(\mathbb{P}_n - \mathbb{P})(\widehat{f} - f)|}{\|\widehat{f} - f\|/\sqrt{n}} \geq t\right\} = \mathbb{P}\left[\mathbb{E}\left\{\frac{|(\mathbb{P}_n - \mathbb{P})(\widehat{f} - f)|}{\|\widehat{f} - f\|/\sqrt{n}} \geq t \;\Big|\; Z^N\right\}\right] \leq \frac{1}{t^2}.$$

Thus for any $\epsilon > 0$ we can pick $t = 1/\sqrt{\epsilon}$ so that the probability above is no more than $\epsilon$, which yields the result. $\square$

These two lemmas allow for proving Theorem 3.1. Recall that this theorem exhibits the empirical mean-like behaviour of doubly robust estimators. This will be key when proving the main result of the paper.

*Proof of Theorem 3.1.* We have that $\hat{\psi}_{\text{DR}} - \psi = Z^* + T_1 + T_2$, where

$$Z^* = (\mathbb{P}_n - \mathbb{P})f, \quad T_1 = (\mathbb{P}_n - \mathbb{P})(\hat{f} - f), \quad T_2 = \mathbb{P}(\hat{f} - f).$$

The consistency of $f$ in $L_2$ norm, together with either the Donsker condition (Lemma C.5) or sample splitting (Lemma C.6), ensures that $T_1 = o_{\mathbb{P}}(1/\sqrt{n})$. For $T_2 = \mathbb{P}(\hat{f} - f)$ and $f = f_1 - f_0$, we have

$$|\mathbb{P}(\hat{f}_1 - f_1)| = |\mathbb{P}\{\frac{A}{\hat{\pi}}(Y - \hat{\theta}_A) + \hat{\theta}_1 - \theta_1\}| = |\mathbb{P}\{(\frac{\pi}{\hat{\pi}} - 1)(\theta_1 - \hat{\theta}_1)\}|$$
$$\leq \frac{1}{\epsilon}\mathbb{P}\{|\pi - \hat{\pi}||\theta_1 - \hat{\theta}_1|\} \leq \frac{1}{\epsilon}\|\pi - \hat{\pi}\|\|\theta_1 - \hat{\theta}_1\|.$$

Same logic applies for $|\mathbb{P}(\hat{f}_0 - f_0)| \leq \frac{1}{1-\epsilon}\|\pi - \hat{\pi}\|\|\theta_0 - \hat{\theta}_0\|$. Therefore $T_2 = o_{\mathbb{P}}(1/\sqrt{n})$ and the result follows. $\square$

We have exhibited the proofs of Lemma C.6 and Theorem 3.1 for sake of completeness. We will extend and prove the analogous results in the $\mathcal{H}$-valued setting in Subsection C.3, which will be needed in demonstrating the main result of the paper. We refer the reader to Kennedy (2022) for a full discussion and exhibition of the aforementioned causal inference results. Lastly, we exhibit the definition of a Glivenko-Cantelli class, which is used in Theorem 4.1.

**Definition C.7** (Glivenko-Cantelli). We say that a class of integrable real-valued functions $\mathcal{F}$ is a Glivenko-Cantelli class for $\mathbb{P}$ if

$$\sup_{f \in \mathcal{F}} \left| \frac{1}{n} \sum_{i=1}^{n} f(X_i) - \mathbb{E}[f(X)] \right|$$

converges to zero in probability as $n \to \infty$.

## C.3 Extension of the results to (infinite-dimensional) vector-valued outcomes

We introduce the extension of Theorem 3.1 to the $\mathcal{H}$-valued outcome setting. For such goal, we also generalize Lemma C.6 to the $\mathcal{H}$-valued scenario and we comment on asymptotically equicontinuous empirical processes Park and Muandet (2023), which will substitute the Donsker class condition in the generalized version of Theorem 3.1.

We start by presenting the generalization of Lemma C.6 to the functional setting.

**Lemma C.8** (Extension of Lemma C.6 to $\mathcal{H}$-valued outcomes). *Let $\mathcal{H}$ be a Hilbert space. Let $\widehat{\omega}(z)$ be a function estimated from a sample $Z^N = (Z_{n+1}, \ldots, Z_N)$ such that $\widehat{\omega}(z) \in \mathcal{H}$ for all $z$, and let $\mathbb{P}_n$ denote the empirical measure over $(Z_1, \ldots, Z_n)$, which is independent of $Z^N$. Then*

$$\|(\mathbb{P}_n - \mathbb{P})(\widehat{\omega} - \omega)\|_{\mathcal{H}} = O_{\mathbb{P}}\left( \frac{\|\widehat{\omega} - \omega\|}{\sqrt{n}} \right).$$

*Proof.* First note that, conditional on $Z^N$, the term in question has mean zero since

$$\begin{aligned}
\mathbb{E}\Big\{ (\mathbb{P}_n - \mathbb{P})(\widehat{\omega} - \omega) \ \Big| \ Z^N \Big\} &= \mathbb{E}\Big\{ \mathbb{P}_n(\widehat{\omega} - \omega) \ \Big| \ Z^N \Big\} - \mathbb{P}(\widehat{\omega} - \omega) \\
&= \mathbb{E}(\widehat{\omega} - \omega \mid Z^N) - \mathbb{P}(\widehat{\omega} - \omega) \\
&= \mathbb{P}(\widehat{\omega} - \omega) - \mathbb{P}(\widehat{\omega} - \omega) \\
&= 0.
\end{aligned}$$

The conditional expectation of the squared norm is

$$\begin{aligned}
\mathbb{E}\Big\{ \|(\mathbb{P}_n - \mathbb{P})(\widehat{\omega} - \omega)\|_{\mathcal{H}}^2 \ \Big| \ Z^N \Big\} &= \mathbb{E}\Big\{ \langle (\mathbb{P}_n - \mathbb{P})(\widehat{\omega} - \omega), (\mathbb{P}_n - \mathbb{P})(\widehat{\omega} - \omega) \rangle_{\mathcal{H}} \ \Big| \ Z^N \Big\} \\
&= \mathbb{E}\Big\{ \langle \frac{1}{n} \sum_{i=1}^{n} [(\widehat{\omega} - \omega)(Z_i) - \mathbb{P}(\widehat{\omega} - \omega)], \frac{1}{n} \sum_{j=1}^{n} [(\widehat{\omega} - \omega)(Z_j) - \mathbb{P}(\widehat{\omega} - \omega)] \rangle_{\mathcal{H}} \ \Big| \ Z^N \Big\} \\
&= \frac{1}{n^2} \sum_{i,j=1}^{n} \mathbb{E}\Big\{ \langle (\widehat{\omega} - \omega)(Z_i) - \mathbb{P}(\widehat{\omega} - \omega), (\widehat{\omega} - \omega)(Z_j) - \mathbb{P}(\widehat{\omega} - \omega) \rangle_{\mathcal{H}} \ \Big| \ Z^N \Big\} \\
&\overset{(i)}{=} \frac{1}{n^2} \sum_{i=1}^{n} \mathbb{E}\Big\{ \langle (\widehat{\omega} - \omega)(Z_i) - \mathbb{P}(\widehat{\omega} - \omega), (\widehat{\omega} - \omega)(Z_i) - \mathbb{P}(\widehat{\omega} - \omega) \rangle_{\mathcal{H}} \ \Big| \ Z^N \Big\} \\
&= \frac{1}{n^2} \sum_{i=1}^{n} \mathbb{E}\Big\{ \|(\widehat{\omega} - \omega)(Z_i) - \mathbb{P}(\widehat{\omega} - \omega)\|_{\mathcal{H}}^2 \ \Big| \ Z^N \Big\} \\
&= \frac{1}{n} \mathbb{E}\Big\{ \|(\widehat{\omega} - \omega)(Z_1) - \mathbb{P}(\widehat{\omega} - \omega)\|_{\mathcal{H}}^2 \ \Big| \ Z^N \Big\} \\
&\overset{(ii)}{=} \frac{1}{n} \Big[ \mathbb{E}\Big\{ \|(\widehat{\omega} - \omega)(Z_1)\|_{\mathcal{H}}^2 \ \Big| \ Z^N \Big\} - \Big\{ \|\mathbb{P}(\widehat{\omega} - \omega)\|_{\mathcal{H}}^2 \ \Big| \ Z^N \Big\} \Big] \\
&= \frac{1}{n} \|\widehat{\omega} - \omega\|^2 - \frac{1}{n} \Big\{ \|\mathbb{P}(\widehat{\omega} - \omega)\|_{\mathcal{H}}^2 \ \Big| \ Z^N \Big\} \\
&\leq \frac{1}{n} \|\widehat{\omega} - \omega\|^2,
\end{aligned}$$

where (i) is obtained given independence of $Z_i, Z_j$ when $i \neq j$ and $\mathbb{E}\left\{ (\mathbb{P}_n - \mathbb{P})(\widehat{\omega} - \omega) \mid Z^N \right\} = 0$, and (ii) by Lemma C.1. Therefore by iterated expectation and Markov's inequality we have

$$
\begin{aligned}
\mathbb{P}\left\{ \frac{\|(\mathbb{P}_n - \mathbb{P})(\widehat{\omega} - \omega)\|_{\mathcal{H}}}{\|\widehat{\omega} - \omega\|/\sqrt{n}} \geq t \right\} &= \mathbb{P}\left[ \mathbb{E}\left\{ \frac{\|(\mathbb{P}_n - \mathbb{P})(\widehat{\omega} - \omega)\|_{\mathcal{H}}}{\|\widehat{\omega} - \omega\|/\sqrt{n}} \geq t \mid Z^N \right\} \right] \\
&= \mathbb{P}\left[ \mathbb{E}\left\{ \frac{\|(\mathbb{P}_n - \mathbb{P})(\widehat{\omega} - \omega)\|_{\mathcal{H}}^2}{\|\widehat{\omega} - \omega\|^2/n} \geq t^2 \mid Z^N \right\} \right] \\
&\leq \frac{1}{t^2}.
\end{aligned}
$$

Thus for any $\epsilon > 0$ we can pick $t = 1/\sqrt{\epsilon}$ so that the probability above is no more than $\epsilon$, which yields the result. $\square$

Note that the former result implies that, if $\|\widehat{\omega} - \omega\| \xrightarrow{p} 0$, then

$$
\|(\mathbb{P}_n - \mathbb{P})(\widehat{\omega} - \omega)\|_{\mathcal{H}} = o_{\mathbb{P}}\left( \frac{1}{\sqrt{n}} \right),
$$

given that we conduct sample splitting. However, sample splitting might not be necessary if our estimators are not flexible enough. Splitting the data might result in a loss in power, hence we are interested in stating another sufficient condition that will lead the *empirical term* $(\mathbb{P}_n - \mathbb{P})(\widehat{\omega} - \omega)$ to be asymptotically negligible without conducting sample splitting. A direct extension of the Donsker class condition stated in Theorem 3.1 to $\mathcal{H}$-valued outcomes is not possible, given that bounding entropies usually make explicit use of the fact that $\mathbb{R}$ is totally ordered (Park and Muandet, 2023). However, Park and Muandet (2023) defines asymptotic equicontinuity, which is precisely the condition which we will require in the extension of Theorem 3.1 to the functional setting.

**Definition C.9** (Asymptotic equicontinuity). We say that the empirical process $\{\nu_n(\omega) = \sqrt{n}(\mathbb{P}_n - \mathbb{P})\omega : \omega \in \mathcal{G}\}$ with values in $\mathcal{H}$ and indexed by $\mathcal{G}$ is *asymptotic equicontinuous* at $\omega_0 \in \mathcal{G}$ if, for every sequence $\{\hat{\omega}_n\} \subset \mathcal{G}$ with $\|\hat{\omega}_n - \omega_0\| \xrightarrow{p} 0$, we have

$$
\|\nu_n(\hat{\omega}_n) - \nu_n(\omega_0)\|_{\mathcal{H}} \xrightarrow{p} 0. \tag{7}
$$

Note that (7) is equivalent to $(\mathbb{P}_n - \mathbb{P})(\widehat{\omega} - \omega) = o_{\mathbb{P}}\left( \frac{1}{\sqrt{n}} \right)$. Park and Muandet (2023) gives sufficient conditions for asymptotic equicontinuity to hold, as well as stating examples of classes that attain asymptotic equicontinuity. We are now ready to present the extension of Theorem 3.1 to the $\mathcal{H}$-valued setting.

**Theorem C.10** (Extension of Theorem 3.1 to $\mathcal{H}$-valued outcomes). *Let* $\phi(z) = \{\frac{a}{\pi(x)} - \frac{1-a}{1-\pi(x)}\}\{y - \beta_a(x)\} + \beta_1(x) - \beta_0(x)$, *where* $y$ *and* $\beta$ *are* $\mathcal{H}$-valued, so that $\Psi = \mathbb{E}\{\phi(Z)\}$ *is the average treatment effect. Suppose that*

- $\hat{\phi}$ *is constructed from an independent sample or the respective empirical process is asymptotically equicontinuous at* $\phi$.

- $\|\hat{\phi} - \phi\| = o_{\mathbb{P}}(1)$.

*Also assume that* $\mathbb{P}(\hat{\pi} \in [\epsilon, 1 - \epsilon]) = 1$. *Then if* $\|\hat{\pi} - \pi\| \sum_a \|\hat{\beta}_a - \beta_a\| = o_{\mathbb{P}}(\frac{1}{\sqrt{n}})$ *it follows that*

$$
\hat{\Psi}_{DR} - \Psi = (\mathbb{P}_n - \mathbb{P})\phi(Z) + \xi_n,
$$

*where* $\|\xi_n\|_{\mathcal{H}} = o_{\mathbb{P}}(\frac{1}{\sqrt{n}})$. *Hence, it is root-n consistent.*

*Proof.* We have that $\hat{\Psi}_{DR} - \Psi = Z^* + T_1 + T_2$, where

$$
Z^* = (\mathbb{P}_n - \mathbb{P})\phi, \quad T_1 = (\mathbb{P}_n - \mathbb{P})(\hat{\phi} - \phi), \quad T_2 = \mathbb{P}(\hat{\phi} - \phi).
$$

The consistency of $\phi$ in $L_2$ norm, together with either asymptotic equicontinuity (Definition C.9) or sample splitting (Lemma C.8), ensures that $\|T_1\|_{\mathcal{H}} = o_{\mathbb{P}}(1/\sqrt{n})$. For $T_2 = \mathbb{P}(\hat{\phi} - \phi)$ and

$\phi = \phi_1 - \phi_0$, we have

$$\|\mathbb{P}(\hat{\phi}_1 - \phi_1)\|_{\mathcal{H}} = \|\mathbb{P}\{\frac{A}{\hat{\pi}}(Y - \hat{\beta}_A) + \hat{\beta}_1 - \beta_1\}\|_{\mathcal{H}} \leq \mathbb{P}\{\|(\frac{\pi}{\hat{\pi}} - 1)(\beta_1 - \hat{\beta}_1)\|_{\mathcal{H}}\}$$

$$\leq \frac{1}{\epsilon}\mathbb{P}\{|\pi - \hat{\pi}|\|\beta_1 - \hat{\beta}_1\|_{\mathcal{H}}\} \leq \frac{1}{\epsilon}\|\pi - \hat{\pi}\|\|\beta_1 - \hat{\beta}_1\|.$$

Same logic applies for $\mathbb{P}(\hat{\phi}_0 - \phi_0) \leq \frac{1}{1-\epsilon}\|\pi - \hat{\pi}\|\|\beta_0 - \hat{\beta}_0\|$. Therefore $\|T_2\|_{\mathcal{H}} = o_{\mathbb{P}}(1/\sqrt{n})$ and the result follows. $\qquad\square$

## C.4 Proof of Theorem 4.1

Assume that we have access to a sample $(X_i, A_i, Y_i)_{i=1}^{2n} \sim (X, A, Y)$. We denote $\mathcal{D}_1 = (X_i, A_i, Y_i)_{i=1}^{n}$, $\mathcal{D}_2 = (X_j, A_j, Y_j)_{j=n+1}^{2n}$. Further,

$$f_h(z) = \frac{1}{n}\sum_{j=n+1}^{2n}\langle\phi(z), \phi(Z_j)\rangle, \quad T_h = \frac{\sqrt{n}\bar{f}_h}{S_h},$$

where $\bar{f}_h$ and $S_h^2$ are the empirical mean and variance respectively:

$$\bar{f}_h = \frac{1}{n}\sum_{i=1}^{n}f_h(Z_i), \quad S_h^2 = \frac{1}{n}\sum_{i=1}^{n}(f_h(Z_i) - \bar{f}_h)^2.$$

In particular, note the indices 1 to $n$ in the definition of $\bar{f}_h$, and the indices $n+1$ to $2n$ when defining $f_h$. Similarly, we denote the analogous version for estimated embeddings $\hat{\phi}$ as follows:

$$f_h^\dagger(z) = \frac{1}{n}\sum_{j=n+1}^{2n}\langle\hat{\phi}^{(1)}(z), \hat{\phi}^{(2)}(Z_j)\rangle, \quad T_h^\dagger = \frac{\sqrt{n}\bar{f}_h^\dagger}{S_h^\dagger}, \quad \bar{f}_h^\dagger = \frac{1}{n}\sum_{i=1}^{n}f_h^\dagger(Z_i), \quad S_h^{\dagger 2} = \frac{1}{n}\sum_{i=1}^{n}(f_h^\dagger(Z_i) - \bar{f}_h^\dagger)^2.$$

For of ease of notation, we also define

$$\tau_1 = \frac{1}{\sqrt{n}}\sum_{i=1}^{n}\phi(Z_i), \quad \tau_2 = \frac{1}{\sqrt{n}}\sum_{j=n+1}^{2n}\phi(Z_j), \quad \hat{\tau}_1 = \frac{1}{\sqrt{n}}\sum_{i=1}^{n}\hat{\phi}^{(1)}(Z_i), \quad \hat{\tau}_2 = \frac{1}{\sqrt{n}}\sum_{j=n+1}^{2n}\hat{\phi}^{(2)}(Z_j).$$

Lastly, note that the following equalities hold by definition:

$$n\bar{f}_h = n\langle\frac{1}{n}\sum_{i=1}^{n}\phi(Z_i), \frac{1}{n}\sum_{j=n+1}^{2n}\phi(Z_j)\rangle = \langle\frac{1}{\sqrt{n}}\sum_{i=1}^{n}\phi(Z_i), \frac{1}{\sqrt{n}}\sum_{j=n+1}^{2n}\phi(Z_j)\rangle = \langle\tau_1, \tau_2\rangle,$$

$$n\bar{f}_h^\dagger = n\langle\frac{1}{n}\sum_{i=1}^{n}\hat{\phi}^{(1)}(Z_i), \frac{1}{n}\sum_{j=n+1}^{2n}\hat{\phi}^{(2)}(Z_j)\rangle = \langle\frac{1}{\sqrt{n}}\sum_{i=1}^{n}\hat{\phi}^{(1)}(Z_i), \frac{1}{\sqrt{n}}\sum_{j=n+1}^{2n}\hat{\phi}^{(2)}(Z_j)\rangle = \langle\hat{\tau}_1, \hat{\tau}_2\rangle,$$

$$(\sqrt{n}S_h)^2 = n\left\{\left(\frac{1}{n}\sum_{i=1}^{n}\langle\phi(Z_i), \frac{1}{n}\sum_{j=n+1}^{2n}\phi(Z_j)\rangle^2\right) - (\bar{f}_h)^2\right\} = \frac{1}{n}\sum_{i=1}^{n}\langle\phi(Z_i), \tau_2\rangle^2 - (\sqrt{n}\bar{f}_h)^2,$$

$$(\sqrt{n}S_h^\dagger)^2 = n\left\{\left(\frac{1}{n}\sum_{i=1}^{n}\langle\hat{\phi}^{(1)}(Z_i), \frac{1}{n}\sum_{j=n+1}^{2n}\hat{\phi}^{(2)}(Z_j)\rangle^2\right) - (\bar{f}_h^\dagger)^2\right\} = \frac{1}{n}\sum_{i=1}^{n}\langle\hat{\phi}^{(1)}(Z_i), \hat{\tau}_2\rangle^2 - (\sqrt{n}\bar{f}_h^\dagger)^2.$$

*Proof of Theorem 4.1.* We will prove the theorem in four steps:

1. We first prove that $n\bar{f}_h^\dagger = n\bar{f}_h + o_{\mathbb{P}}(1)$.

2. We then show that $nS_h^{\dagger 2} = nS_h^2 + o_{\mathbb{P}}(1)$.

3. Further, we prove that $\frac{1}{\mathbb{E}[n\{f_h(Z)\}^2|\mathcal{D}_2]} = O_{\mathbb{P}}(1)$.

4. We conclude that $T_h^\dagger \xrightarrow{d} N(0, 1)$.

It now remains to prove each of the four steps.

**Details of step 1.** Theorem C.10 may be applied for both $\hat{\phi}^{(0)}, \hat{\phi}^{(1)}$ because its conditions are fulfilled. We obtain that

$$\hat{\tau}_1 = \tau_1 + \sqrt{n}\xi_n^{(1)}, \quad \hat{\tau}_2 = \tau_2 + \sqrt{n}\xi_n^{(2)},$$

where $\|\sqrt{n}\xi_n^{(1)}\|_{\mathcal{H}}, \|\sqrt{n}\xi_n^{(2)}\|_{\mathcal{H}} = o_{\mathbb{P}}(1)$. Rewriting the expression, we note that

$$\hat{\tau}_1 = \tau_1 + \chi_n^{(1)}, \quad \hat{\tau}_2 = \tau_2 + \chi_n^{(2)},$$

where $\|\chi_n^{(1)}\|_{\mathcal{H}}, \|\chi_n^{(2)}\|_{\mathcal{H}} = o_{\mathbb{P}}(1)$. Based on Theorem C.3,

$$\tau_1 \xrightarrow{d} N(0, C), \quad \tau_2 \xrightarrow{d} N(0, C),$$

where $C = \mathbb{E}[\phi(Z) \otimes \phi(Z)]$, which implies $\max(\|\tau_1\|_{\mathcal{H}}^2, \|\tau_2\|_{\mathcal{H}}^2) = O_{\mathbb{P}}(1)$. Consequently,

$$
\begin{aligned}
n\bar{f}_h^{\dagger} &= \langle \hat{\tau}_1, \hat{\tau}_2 \rangle \\
&= \langle \tau_1 + \chi_n^{(1)}, \tau_2 + \chi_n^{(2)} \rangle \\
&= \langle \tau_1, \tau_2 \rangle + \langle \tau_1, \chi_n^{(2)} \rangle + \langle \chi_n^{(1)}, \tau_2 \rangle + \langle \chi_n^{(1)}, \chi_n^{(2)} \rangle \\
&= n\bar{f}_h + \langle \tau_1, \chi_n^{(2)} \rangle + \langle \chi_n^{(1)}, \tau_2 \rangle + \langle \chi_n^{(1)}, \chi_n^{(2)} \rangle.
\end{aligned}
$$

Given that $|\langle \tau_1, \chi_n^{(2)} \rangle + \langle \chi_n^{(1)}, \tau_2 \rangle + \langle \chi_n^{(1)}, \chi_n^{(2)} \rangle|$ is upper bounded by $\|\tau_1\|_{\mathcal{H}}\|\chi_n^{(2)}\|_{\mathcal{H}} + \|\chi_n^{(1)}\|_{\mathcal{H}}\|\tau_2\|_{\mathcal{H}} + \|\chi_n^{(1)}\|_{\mathcal{H}}\|\chi_n^{(2)}\|_{\mathcal{H}}$ (by the triangle inequality and Cauchy-Schwarz inequality), which is $O_{\mathbb{P}}(1)o_{\mathbb{P}}(1) + o_{\mathbb{P}}(1)O_{\mathbb{P}}(1) + o_{\mathbb{P}}(1)o_{\mathbb{P}}(1) = o_{\mathbb{P}}(1)$, we deduce that

$$n\bar{f}_h^{\dagger} = n\bar{f}_h + o_{\mathbb{P}}(1). \tag{8}$$

**Details of step 2.** We claim that

$$\left| \frac{1}{n}\sum_{i=1}^{n} \langle \hat{\phi}^{(1)}(Z_i), \hat{\tau}_2 \rangle^2 - \frac{1}{n}\sum_{i=1}^{n} \langle \phi(Z_i), \tau_2 \rangle^2 \right| = o_{\mathbb{P}}(1), \tag{9}$$

as well as

$$(\sqrt{n}\bar{f}_h^{\dagger})^2 = (\sqrt{n}\bar{f}_h)^2 + o_{\mathbb{P}}(1). \tag{10}$$

If both (9) and (10) hold, we obtain

$$
\begin{aligned}
(\sqrt{n}S_h^{\dagger})^2 - (\sqrt{n}S_h)^2 &= \left( \frac{1}{n}\sum_{i=1}^{n} \langle \hat{\phi}^{(1)}(Z_i), \hat{\tau}_2 \rangle^2 - (\sqrt{n}\bar{f}_h^{\dagger})^2 \right) - \left( \frac{1}{n}\sum_{i=1}^{n} \langle \phi(Z_i), \tau_2 \rangle^2 - (\sqrt{n}\bar{f}_h)^2 \right) \\
&= \left( \frac{1}{n}\sum_{i=1}^{n} \langle \hat{\phi}^{(1)}(Z_i), \hat{\tau}_2 \rangle^2 - \frac{1}{n}\sum_{i=1}^{n} \langle \phi(Z_i), \tau_2 \rangle^2 \right) - \left( (\sqrt{n}\bar{f}_h^{\dagger})^2 - (\sqrt{n}\bar{f}_h)^2 \right) \\
&= o_{\mathbb{P}}(1) + o_{\mathbb{P}}(1) \\
&= o_{\mathbb{P}}(1),
\end{aligned}
$$

hence

$$nS_h^{\dagger^2} = nS_h^2 + o_{\mathbb{P}}(1), \tag{11}$$

thus concluding step 2.

In order to prove (9), we denote $\epsilon_i = \hat{\phi}^{(1)}(Z_i) - \phi(Z_i)$ and consider

$$
\begin{aligned}
|\frac{1}{n}\sum_{i=1}^{n}\langle\hat{\phi}^{(1)}(Z_i),\tau_2\rangle^2 - \frac{1}{n}\sum_{i=1}^{n}\langle\phi(Z_i),\tau_2\rangle^2| &= |\frac{1}{n}\sum_{i=1}^{n}\langle\phi(Z_i)+\epsilon_i,\tau_2\rangle^2 - \frac{1}{n}\sum_{i=1}^{n}\langle\phi(Z_i),\tau_2\rangle^2| \\
&= |\frac{1}{n}\sum_{i=1}^{n}\langle\epsilon_i,\tau_2\rangle^2 + \frac{2}{n}\sum_{i=1}^{n}\langle\phi(Z_i),\tau_2\rangle\langle\epsilon_i,\tau_2\rangle| \\
&\overset{(i)}{\leq} |\frac{1}{n}\sum_{i=1}^{n}\langle\epsilon_i,\tau_2\rangle^2| + |\frac{2}{n}\sum_{i=1}^{n}\langle\phi(Z_i),\tau_2\rangle\langle\epsilon_i,\tau_2\rangle| \\
&\overset{(ii)}{\leq} \frac{1}{n}\sum_{i=1}^{n}\langle\epsilon_i,\tau_2\rangle^2 + 2\left(\frac{1}{n}\sum_{i=1}^{n}\langle\phi(Z_i),\tau_2\rangle^2\right)^{\frac{1}{2}}\left(\frac{1}{n}\sum_{i=1}^{n}\langle\epsilon_i,\tau_2\rangle^2\right)^{\frac{1}{2}} \\
&\overset{(iii)}{\leq} \frac{1}{n}\sum_{i=1}^{n}\langle\epsilon_i,\tau_2\rangle^2 + 2\left(\frac{1}{n}\sum_{i=1}^{n}\|\phi(Z_i)\|_{\mathcal{H}}^2\|\tau_2\|_{\mathcal{H}}^2\right)^{\frac{1}{2}}\left(\frac{1}{n}\sum_{i=1}^{n}\langle\epsilon_i,\tau_2\rangle^2\right)^{\frac{1}{2}} \\
&\overset{(iv)}{\leq} \frac{1}{n}\sum_{i=1}^{n}\|\epsilon_i\|_{\mathcal{H}}^2\|\tau_2\|_{\mathcal{H}}^2 + \\
&\quad + 2\left(\frac{1}{n}\sum_{i=1}^{n}\|\phi(Z_i)\|_{\mathcal{H}}^2\|\tau_2\|_{\mathcal{H}}^2\right)^{\frac{1}{2}}\left(\frac{1}{n}\sum_{i=1}^{n}\|\epsilon_i\|_{\mathcal{H}}^2\|\tau_2\|_{\mathcal{H}}^2\right)^{\frac{1}{2}},
\end{aligned}
$$

where (i) is obtained by the triangle inequality, and (ii), (iii), (iv) are obtained by Cauchy-Schwarz inequality. We recall that $\|\tau_2\|_{\mathcal{H}}^2 = O_{\mathbb{P}}(1)$. Moreover, $\mathbb{E}\|\phi(Z)\|_{\mathcal{H}}^2 \leq \mathbb{E}\|\phi(Z)\|_{\mathcal{H}}^4 < \infty$ implies $\frac{1}{n}\sum_{i=1}^{n}\|\phi(Z_i)\|_{\mathcal{H}}^2 \overset{p}{\to} \mathbb{E}\|\phi(Z)\|_{\mathcal{H}}^2$ by the law of large numbers, so $\frac{1}{n}\sum_{i=1}^{n}\|\phi(Z_i)\|_{\mathcal{H}}^2 = O_{\mathbb{P}}(1)$.

If $\hat{\phi}$ was constructed independently from $\mathcal{D}_1$, then

$$
\mathbb{P}\left[\frac{1}{n}\sum_{i=1}^{n}\|\epsilon_i\|_{\mathcal{H}}^2\right] = \mathbb{P}\left[\|\epsilon_1\|_{\mathcal{H}}^2\right] = \mathbb{P}\left[\|\hat{\phi}^{(1)}-\phi\|_{\mathcal{H}}^2\right] = \|\hat{\phi}^{(1)}-\phi\|^2 = o_{\mathbb{P}}(1),
$$

so by Markov's inequality

$$
\frac{1}{n}\sum_{i=1}^{n}\|\epsilon_i\|_{\mathcal{H}}^2 = o_{\mathbb{P}}(1).
$$

If $\|\hat{\phi}^{(1)}\|_{\mathcal{H}}^2$ belongs to a Glivenko-Cantelli class, then $\|\hat{\phi}^{(1)}-\phi\|_{\mathcal{H}}^2$ belongs to a Glivenko-Cantelli class. From this, we conclude that $(\mathbb{P}-\mathbb{P}_n)\|\hat{\phi}^{(1)}-\phi\|_{\mathcal{H}}^2 = o_{\mathbb{P}}(1)$, hence

$$
\begin{aligned}
\frac{1}{n}\sum_{i=1}^{n}\|\epsilon_i\|_{\mathcal{H}}^2 &= \mathbb{P}_n\|\hat{\phi}^{(1)}-\phi\|_{\mathcal{H}}^2 \\
&= (\mathbb{P}-\mathbb{P}_n)\|\hat{\phi}^{(1)}-\phi\|_{\mathcal{H}}^2 + \mathbb{P}\|\hat{\phi}^{(1)}-\phi\|_{\mathcal{H}}^2 \\
&= (\mathbb{P}-\mathbb{P}_n)\|\hat{\phi}^{(1)}-\phi\|_{\mathcal{H}}^2 + \|\hat{\phi}^{(1)}-\phi\|^2 \\
&= o_{\mathbb{P}}(1).
\end{aligned}
$$

In either case,

$$\left| \frac{1}{n} \sum_{i=1}^{n} \langle \hat{\phi}^{(1)}(Z_i), \tau_2 \rangle^2 - \frac{1}{n} \sum_{i=1}^{n} \langle \phi(Z_i), \tau_2 \rangle^2 \right| \leq \frac{1}{n} \sum_{i=1}^{n} \|\epsilon_i\|_{\mathcal{H}}^2 \|\tau_2\|_{\mathcal{H}}^2 +$$

$$+ 2 \left( \frac{1}{n} \sum_{i=1}^{n} \|\phi(Z_i)\|_{\mathcal{H}}^2 \|\tau_2\|_{\mathcal{H}}^2 \right)^{\frac{1}{2}} \left( \frac{1}{n} \sum_{i=1}^{n} \|\epsilon_i\|_{\mathcal{H}}^2 \|\tau_2\|_{\mathcal{H}}^2 \right)^{\frac{1}{2}}$$

$$\leq \|\tau_2\|_{\mathcal{H}}^2 \left( \frac{1}{n} \sum_{i=1}^{n} \|\epsilon_i\|_{\mathcal{H}}^2 \right) +$$

$$+ 2\|\tau_2\|_{\mathcal{H}}^2 \left( \frac{1}{n} \sum_{i=1}^{n} \|\phi(Z_i)\|_{\mathcal{H}}^2 \right)^{\frac{1}{2}} \left( \frac{1}{n} \sum_{i=1}^{n} \|\epsilon_i\|_{\mathcal{H}}^2 \right)^{\frac{1}{2}}$$

$$= O_{\mathbb{P}}(1) o_{\mathbb{P}}(1) + O_{\mathbb{P}}(1) O_{\mathbb{P}}(1) o_{\mathbb{P}}(1)$$

$$= o_{\mathbb{P}}(1).$$

Further,

$$\left| \frac{1}{n} \sum_{i=1}^{n} \langle \hat{\phi}^{(1)}(Z_i), \hat{\tau}_2 \rangle^2 - \frac{1}{n} \sum_{i=1}^{n} \langle \hat{\phi}^{(1)}(Z_i), \tau_2 \rangle^2 \right| = \left| \frac{1}{n} \sum_{i=1}^{n} \langle \hat{\phi}^{(1)}(Z_i), \tau_2 + \chi_n^{(2)} \rangle^2 - \frac{1}{n} \sum_{i=1}^{n} \langle \hat{\phi}^{(1)}(Z_i), \tau_2 \rangle^2 \right|$$

$$= \left| \frac{1}{n} \sum_{i=1}^{n} \langle \hat{\phi}^{(1)}(Z_i), \chi_n^{(2)} \rangle^2 + \frac{2}{n} \sum_{i=1}^{n} \langle \hat{\phi}^{(1)}(Z_i), \tau_2 \rangle \langle \hat{\phi}^{(1)}(Z_i), \chi_n^{(2)} \rangle \right|$$

$$\overset{(i)}{\leq} \left| \frac{1}{n} \sum_{i=1}^{n} \langle \hat{\phi}^{(1)}(Z_i), \chi_n^{(2)} \rangle^2 \right| + \left| \frac{2}{n} \sum_{i=1}^{n} \langle \hat{\phi}^{(1)}(Z_i), \tau_2 \rangle \langle \hat{\phi}^{(1)}(Z_i), \chi_n^{(2)} \rangle \right|$$

$$\overset{(ii)}{\leq} \frac{1}{n} \sum_{i=1}^{n} \langle \hat{\phi}^{(1)}(Z_i), \chi_n^{(2)} \rangle^2 +$$

$$+ 2 \left( \frac{1}{n} \sum_{i=1}^{n} \langle \hat{\phi}^{(1)}(Z_i), \tau_2 \rangle^2 \right)^{\frac{1}{2}} \left( \frac{1}{n} \sum_{i=1}^{n} \langle \hat{\phi}^{(1)}(Z_i), \chi_n^{(2)} \rangle^2 \right)^{\frac{1}{2}}$$

$$\overset{(iii)}{\leq} \left( \frac{1}{n} \sum_{i=1}^{n} \|\hat{\phi}^{(1)}(Z_i)\|_{\mathcal{H}}^2 \right) \left( \|\chi_n^{(2)}\|_{\mathcal{H}}^2 + 2\|\tau_2\|_{\mathcal{H}} \|\chi_n^{(2)}\|_{\mathcal{H}} \right)$$

$$= O_{\mathbb{P}}(1) \{ o_{\mathbb{P}}(1) + O_{\mathbb{P}}(1) o_{\mathbb{P}}(1) \}$$

$$= o_{\mathbb{P}}(1),$$

where we obtain (i) by the triangle inequality, (ii) by Cauchy-Schwarz inequality, and (iii) by Cauchy-Schwarz inequality and grouping terms. By the triangle inequality, $|\frac{1}{n} \sum_{i=1}^{n} \langle \hat{\phi}^{(1)}(Z_i), \hat{\tau}_2 \rangle^2 - \frac{1}{n} \sum_{i=1}^{n} \langle \phi(Z_i), \tau_2 \rangle^2|$ is upper bounded by

$$\left| \frac{1}{n} \sum_{i=1}^{n} \langle \hat{\phi}^{(1)}(Z_i), \hat{\tau}_2 \rangle^2 - \frac{1}{n} \sum_{i=1}^{n} \langle \hat{\phi}^{(1)}(Z_i), \tau_2 \rangle^2 \right| + \left| \frac{1}{n} \sum_{i=1}^{n} \langle \hat{\phi}^{(1)}(Z_i), \tau_2 \rangle^2 - \frac{1}{n} \sum_{i=1}^{n} \langle \phi(Z_i), \tau_2 \rangle^2 \right|,$$

which is $= o_{\mathbb{P}}(1) + o_{\mathbb{P}}(1) = o_{\mathbb{P}}(1)$, thus (9) holds.

In order to prove (10), we note that (8) implies $(n\bar{f}_h^\dagger)^2 = (n\bar{f}_h + o_{\mathbb{P}}(1))^2$. Expanding the second term we obtain

$$(n\bar{f}_h + o_{\mathbb{P}}(1))^2 = (n\bar{f}_h)^2 + (o_{\mathbb{P}}(1))^2 + (n\bar{f}_h) o_{\mathbb{P}}(1)$$

$$= (n\bar{f}_h)^2 + (o_{\mathbb{P}}(1))^2 + O_{\mathbb{P}}(1) o_{\mathbb{P}}(1)$$

$$= (n\bar{f}_h)^2 + o_{\mathbb{P}}(1),$$

given that $|n\bar{f}_h| = |\langle \tau_1, \tau_2 \rangle| \leq \|\tau_1\|_{\mathcal{H}} \|\tau_2\|_{\mathcal{H}} = O_{\mathbb{P}}(1) O_{\mathbb{P}}(1) = O_{\mathbb{P}}(1)$. Consequently, $(n\bar{f}_h^\dagger)^2 = (n\bar{f}_h)^2 + o_{\mathbb{P}}(1)$, which implies (10) dividing by $n$ in both sides.

**Details of step 3.** We claim that

$$\frac{1}{\mathbb{E}[n\{f_h(Z)\}^2|\mathcal{D}_2]} = O_{\mathbb{P}}(1). \tag{12}$$

To show this, we note that

$$\mathbb{E}[n\{f_h(Z)\}^2|\mathcal{D}_2] = \mathbb{E}[\langle\phi(Z),\tau_2\rangle^2|\mathcal{D}_2] \stackrel{(i)}{=} \langle C\tau_2,\tau_2\rangle = \sum_{i=1}^{\infty}\lambda_j\beta_j^2,$$

where

- $C = \mathbb{E}[\phi(Z)\otimes\phi(Z)]$, $(\lambda_j, j \geq 1)$ is the sequence of non-negative eigenvalues in decreasing order of the covariance operator $C$,

- $(v_j, j \geq 1)$ is the sequence of orthonormal eigenfunctions of the covariance operator $C$ (and basis of $\mathcal{H}$ as well),

- $\tau_2 = \sum_{i=j}^{\infty}\beta_j v_j$, being $(\beta_j, j \geq 1)$ the random coefficients of $\tau_2$ with respect to the basis $(v_j, j \geq 1)$,

and (i) is obtained from Lemma C.2 and the fact that $\mathbb{E}[\phi(Z)] = 0$, $\mathbb{E}[\|\phi(Z)\|_{\mathcal{H}}^2] \leq \mathbb{E}[\|\phi(Z)\|_{\mathcal{H}}^4] < \infty$. Based on Lemma C.4,

$$\tau_2 \stackrel{d}{\to} \sum_{j=1}^{\infty}\sqrt{\lambda_j}N_j v_j,$$

so by the continuous mapping theorem for random variables in separable Banach spaces (Bosq, 2000, Equation 2.11)

$$\beta_1 = \langle\tau_2, v_1\rangle \stackrel{d}{\to} \langle\sum_{j=1}^{\infty}\sqrt{\lambda_j}N_j v_j, v_1\rangle = \sqrt{\lambda_1}N_1.$$

Consequently,

$$\mathbb{E}[n\{f_h(Z)\}^2|\mathcal{D}_2] = \sum_{i=1}^{\infty}\lambda_j\beta_j^2 \geq \lambda_1\beta_1^2 \stackrel{d}{\to} \lambda_1^2 N_1^2. \tag{13}$$

Further, by Cauchy-Schwarz inequality,

$$0 < \mathbb{E}\left[\langle\phi(Z_1),\phi(Z_2)\rangle^2\right] \leq \mathbb{E}\left[\|\phi(Z_1)\|_{\mathcal{H}}^2\|\phi(Z_2)\|_{\mathcal{H}}^2\right] = \mathbb{E}^2\left[\|\phi(Z)\|_{\mathcal{H}}^2\right]$$

$$= \left(\mathbb{E}\left[\sum_{j=1}^{\infty}\langle\phi(Z),v_j\rangle^2\right]\right)^2$$

$$= \left(\sum_{j=1}^{\infty}\mathbb{E}\langle\phi(Z),v_j\rangle^2\right)^2$$

$$= \left(\sum_{j=1}^{\infty}\langle Cv_j,v_j\rangle\right)^2$$

$$= \left(\sum_{j=1}^{\infty}\lambda_j\right)^2.$$

That is, the sum of non-negative eigenvalues is strictly greater than zero, so at least one of them has to be strictly positive. By the non-increasing order of the eigenvalues, this implies that the first one is strictly positive i.e., $\lambda_1 > 0$. For any $M > 0$ it holds that

$$\mathbb{P}\left(\mathbb{E}[n\{f_h(Z)\}^2|\mathcal{D}_2] \leq \frac{1}{M}\right) = \mathbb{P}\left(\sum_{i=1}^{\infty}\lambda_j\beta_j^2 \leq \frac{1}{M}\right) \leq \mathbb{P}\left(\lambda_1\beta_1^2 \leq \frac{1}{M}\right).$$

Given $\lambda_1^2 > 0$ and (13), we also obtain

$$\mathbb{P}\left(\lambda_1\beta_1^2 \le \frac{1}{M}\right) \overset{n\to\infty}{\rightsquigarrow} \mathbb{P}\left(N_1^2 \le \frac{1}{M\lambda_1^2}\right) = F\left(\frac{1}{M\lambda_1^2}\right)$$

where $F$ is the cdf of a $\chi_1^2$ distribution. Hence for every $\epsilon > 0$, there exists $m \in \mathbb{N}$ such that

$$\left|\mathbb{P}\left(\lambda_1^2\beta_1^2 \le \frac{1}{M}\right) - F\left(\frac{1}{M\lambda_1^2}\right)\right| < \frac{\epsilon}{2}$$

for $n \ge m$, which implies

$$\mathbb{P}\left(\lambda_1^2\beta_1^2 \le \frac{1}{M}\right) < \frac{\epsilon}{2} + F\left(\frac{1}{M\lambda_1^2}\right).$$

Further, $\lim_{w\to 0} F(w) = 0$, so there exists $M^* > 0$ such that $F\left(\frac{1}{M^*\lambda_1^2}\right) < \frac{\epsilon}{2}$. Consequently, for every $\epsilon > 0$, there exist $M^* > 0$ and $m \in \mathbb{N}$ such that

$$\mathbb{P}\left(\lambda_1^2\beta_1^2 \le \frac{1}{M^*}\right) < \frac{\epsilon}{2} + \frac{\epsilon}{2} = \epsilon,$$

for $n \ge m$, so

$$\mathbb{P}\left(\frac{1}{\mathbb{E}[n\{f_h(Z)\}^2|\mathcal{D}_2]} \ge M^*\right) = \mathbb{P}\left(\mathbb{E}[n\{f_h(Z)\}^2|\mathcal{D}_2] \le \frac{1}{M}\right) \le \mathbb{P}\left(\lambda_1^2\beta_1^2 \le \frac{1}{M^*}\right) < \epsilon,$$

thus concluding (12).

**Details of step 4.** Given Conditions (i), (ii), and (iii), Kim and Ramdas (2023, Proof of Theorem 4.2) proved that

$$\frac{n\bar{f}_h}{\sqrt{\mathbb{E}[n\{f_h(Z)\}^2|\mathcal{D}_2]}} = \frac{\sqrt{n}\bar{f}_h}{\sqrt{\mathbb{E}[f_h^2(Z)|\mathcal{D}_2]}} \overset{d}{\to} N(0,1). \tag{14}$$

as well as

$$\frac{n(S_h)^2}{\mathbb{E}[n\{f_h(Z)\}^2|\mathcal{D}_2]} = \frac{S_h^2}{\mathbb{E}[f_h^2(Z)|\mathcal{D}_2]} \overset{p}{\to} 1. \tag{15}$$

Combining (8) and (12), we obtain

$$\frac{n\bar{f}_h^\dagger}{\sqrt{\mathbb{E}[n\{f_h(Z)\}^2|\mathcal{D}_2]}} = \frac{n\bar{f}_h + o_\mathbb{P}(1)}{\sqrt{\mathbb{E}[n\{f_h(Z)\}^2|\mathcal{D}_2]}}$$

$$= \frac{n\bar{f}_h}{\sqrt{\mathbb{E}[n\{f_h(Z)\}^2|\mathcal{D}_2]}} + o_\mathbb{P}(1)O_\mathbb{P}(1)$$

$$= \frac{n\bar{f}_h}{\sqrt{\mathbb{E}[n\{f_h(Z)\}^2|\mathcal{D}_2]}} + o_\mathbb{P}(1),$$

and based on (14) and Slutsky's theorem,

$$\frac{n\bar{f}_h^\dagger}{\sqrt{\mathbb{E}[n\{f_h(Z)\}^2|\mathcal{D}_2]}} \overset{d}{\to} N(0,1). \tag{16}$$

Combining (11) and (12), we obtain

$$\frac{nS_h^{\dagger 2}}{\mathbb{E}[n\{f_h(Z)\}^2|\mathcal{D}_2]} = \frac{nS_h^2 + o_\mathbb{P}(1)}{\mathbb{E}[n\{f_h(Z)\}^2|\mathcal{D}_2]}$$

$$= \frac{nS_h^2}{\mathbb{E}[n\{f_h(Z)\}^2|\mathcal{D}_2]} + o_\mathbb{P}(1)O_\mathbb{P}(1)$$

$$= \frac{nS_h^2}{\mathbb{E}[n\{f_h(Z)\}^2|\mathcal{D}_2]} + o_\mathbb{P}(1).$$

Thus based on (15), we obtain

$$\frac{n S_h^{\dagger\,2}}{\mathbb{E}[n\{f_h(Z)\}^2|\mathcal{D}_2]} \xrightarrow{p} 1,$$

and by the continuous mapping theorem

$$\frac{\sqrt{\mathbb{E}[n\{f_h(Z)\}^2|\mathcal{D}_2]}}{\sqrt{n} S_h^{\dagger}} \xrightarrow{p} 1. \tag{17}$$

Combining (16) and (17) and based on Slutsky's theorem, we obtain

$$\frac{\sqrt{n}\bar{f}_h^{\dagger}}{S_h^{\dagger}} = \underbrace{\frac{n\bar{f}_h^{\dagger}}{\sqrt{\mathbb{E}[n\{f_h(Z)\}^2|\mathcal{D}_2]}}}_{\xrightarrow{d} N(0,1)} \underbrace{\frac{\sqrt{\mathbb{E}[n\{f_h(Z)\}^2|\mathcal{D}_2]}}{\sqrt{n} S_h^{\dagger}}}_{\xrightarrow{p} 1} \xrightarrow{d} N(0,1).$$

Hence

$$T_h^{\dagger} = \frac{\sqrt{n}\bar{f}_h^{\dagger}}{S_h^{\dagger}} \xrightarrow{d} N(0,1).$$

$\square$

