# OpenReview forum: "An Efficient Doubly-Robust Test for the Kernel Treatment Effect"
_NeurIPS.cc/2023/Conference — NeurIPS 2023 poster_

### Official Review · Reviewer_DRRe · 2023-07-04

**Soundness:** 4 excellent
**Presentation:** 4 excellent
**Contribution:** 4 excellent
**Rating:** 7
**Confidence:** 4

**Summary:**

This paper proposes Augmented Inverse Propensity Weighted cross Kernel Treatment Test (AIPW-xKTE), which is a doubly robust test with provably valid type-I error based on kernel mean embeddings to test for distributional treatment effect. The paper has one result, Theorem 4.1, showing the asymptotic normality of the proposed test statistic, and demonstrates its performance on synthetic and real datasets.

**Strengths:**

The paper has one clear goal, i.e. to provide a testing procedure for distributional treatment effect. As the authors mention, distributional treatment effect has been an important topic of research for a while, and their proposal, AIPW-xKTE, has several advantages over the previous methods, most prominently that it has an analytical asymptotic null distribution, circumventing the need for permutation to get the null distribution, as well as being doubly robust.

The paper is very clearly written, setting out its goal in the backdrop of previous works and carrying out that goal with minimal fuss. The paper only offers one result, Theorem 4.1, but I think its conciseness is its value. I haven't gone through every detail of the proof in the appendix but a scan convinced me of its soundness. The paper was a pleasure to read.

**Weaknesses:**

Perhaps one thing to count against this paper is its lack of novelty, in that its proposal is not something completely novel, rather it combines two ideas, namely cross U-statistics and augmented inverse probability weighting. However, it combines them to good effect and conducts thorough analysis of it, both theoretically and empirically, and I do not think this should count heavily against the paper.

**Questions:**

The bottleneck of this procedure would probably be the estimation of kernel conditional mean embeddings, which has $n^3$ complexity? Perhaps it would be worth looking at speeding this up, through approximate kernel ridge regression methods (e.g. Nystrom method proposed in [Grunewalder et al., 2012] or FALKON in [Rudi et al., 2017]).

**Limitations:**

The conclusion section has listed a few of its limitations and interesting possible directions for future research.

---

> ### Author Rebuttal · Authors · 2023-08-08
>
> We thank the reviewer for the comments.
>
> - The bottleneck of this procedure would probably be the estimation of kernel conditional mean embeddings, which has n3 complexity? Perhaps it would be worth looking at speeding this up, through approximate kernel ridge regression methods.
>
> We agree that looking at approximate kernel ridge regression methods for faster approximation of kernel conditional mean embeddings would be an interesting point to consider in future work.

---

> > ### Comment · Reviewer_DRRe · 2023-08-13
> > **Thank you**
> >
> > I have no more comments to make, and would like to keep my evaluation of the paper. Good luck!

---

> > > ### Author Response · Authors · 2023-08-18
> > >
> > > We would like to thank the reviewer again for their time and comments.

---

### Official Review · Reviewer_NqCE · 2023-07-05

**Soundness:** 4 excellent
**Presentation:** 4 excellent
**Contribution:** 3 good
**Rating:** 8
**Confidence:** 5

**Summary:**

The paper proposes a test of the null hypothesis that a binary treatment has no effect on the the potential outcome distribution. The test combines ideas of kernel mean embedding, double robustness, and cross U-statistics.

**Strengths:**

Originality

-The connection between kernel embeddings of effect distributions and the cross U-statistic appears to be new.

-The main difference from Shekhar et al. (2022), who combine kernel mean embedding and cross U-statistics, appears to be the connection to double robustness.

-The main difference from Fawkes et al (2022), who combine kernel mean embedding and double robustness, appears to be the connection to cross U-statistics.

-For the kernel embedding of the potential outcome distribution, Muandet et al. (2021) use an IPW-style estimator while Singh et al. (2020) use a regression-style estimator. Similar to Fawkes et al. (2022), this paper combines IPW and regression estimators into a doubly robust AIPW estimator.

Quality - The results are clear and appear to be correct, with some minor comments given below.

Clarity - The paper is well written, especially its appendix.

Significance - Ultimately, this is a paper that combines building blocks that have been partially combined before. The combination is well executed, and contributes to the literature.

**Weaknesses:**

I will raise the score if these items are addressed.



Statistical concepts

-The paper advertises efficiency, which when discussing tests, refers to certain statistical properties. However, the efficiency being described is computational by avoiding permutations. The framing should clarify this.

-Asymptotic equicontinuity and Glivenko-Cantelli class are not well explained in the main text. The former is well explained in the appendix, so a pointer would suffice. The latter is not; please provide more explanation of what this condition means and why it is reasonable in this context.

-Some statements about the asymptotic variance are too strong or poorly worded: on line 19 “the asymptotic variance…” and line  164 “the asymptotic variance…”

Comparisons

-The references given for average treatment effect are actually for the local average treatment effect on line 21. Please update here and elsewhere.

-The doubly robust kernel mean embedding estimator can be viewed as augmenting IPW (Muandet et al. 2021) with regression (Singh et al. 2020) approaches to kernel mean embeddings of potential outcome distributions, just as AIPW augments IPW with regression approaches to treatment effects. It would be worthwhile to point this out.

-It would be good to see brief comparisons to Shekhar et al. (2022) and Fawkes et al (2022) following Theorem 4.1.

Notation

-Sometimes notation is overloaded, which is unnecessary and a bit confusing. For example, mu refers to a kernel mean embedding, a regression, and something else in the appendix.

-Another notation issue is that the norm for beta is not defined in Theorem 4.1, and the cross fitting is poorly explained compared to Algorithm 1.

-It is not a good notation choice to write k(w,y) when x and y are variables with specific meanings in the paper.

-Finally, replace O(100n^2) with O(Bn^2).

**Questions:**

The authors write “We were unable to control the type 1 error of the test presented in Fawkes (2022)…” What does this mean? Why not include Fawkes (2022), the most closely related work, in the simulations?

The authors show computational efficiency, but how about statistical efficiency?

In inequality (iii) of line 656, shouldn’t there be a 2 on the last term?

I had some issues with the proof of Step 3. Shouldn’t the final expression have lambda_1^2 in (13)?  This correction would continue throughout the proof. At the bottom of page 27, how does the previous display imply lambda_1>0?

**Limitations:**

There are no issues.

---

> ### Author Rebuttal · Authors · 2023-08-08
>
> We thank the reviewer for the comments. We would like to address the following weaknesses and questions raised by the reviewer.
> - 1
>
> We have replaced the word "efficient" by "computationally efficient" in lines 7 and 307 to clarify that we are referring to computational efficiency.
> - 2
>
> We have replaced lines 190-191 by "…we ought to refer to asymptotically equicontinuous empirical processes (Park and Muandet, 2023) and Glivenko-Cantelli classes. We refer the reader to Appendix C for a presentation of such concepts, clarification of the norms used, and the proof of the following theorem."
> and we have included the following definition of a Glivenko-Cantelli class in the same appendix:
>
> Definition (Glivenko-Cantelli). We say that a class of integrable real-valued functions \mathcal{F} is a Glivenko-Cantelli class for \mathbb{P} if
> \begin{equation*}
> sup_{f \in \mathcal{F}} \| \frac{1}{n} \sum_{i=1}^n f(X_i) - \mathbb{E}[f(X)] \|
> \end{equation*}
> converges to zero in probability as $n \to \infty$.
>
>
> We highlight that the Glivenko-Cantelli condition is required on the squared norm of the estimator (line 195), which is a scalar, hence we are dealing with the classical Glivenko-Cantelli concept. This condition appears to be a light assumption, as it is frequently satisfied by parametric function classes in low dimensional settings, and we are imposing it on the norm of the estimator, hence retrieving a one-dimensional setting. It is used to control the asymptotic behavior of the residuals in line 654.
> - 3
>
> We have replaced line 29 by "Under certain conditions (e.g., consistent nuisance estimation at $n^{-1/4}$ rates), the asymptotic mean squared error of the AIPW estimator is smaller than that of the IPW and PI estimator, and minimax optimal in a local asymptotic sense (Kennedy, 2020), hence... ", with the following cite:
> Kennedy, Edward H. "Semiparametric doubly robust targeted double machine learning: a review." arXiv preprint arXiv:2203.06469 (2022).
>
> We have replaced lines 164-166 by "Under certain conditions (e.g., consistent nuisance estimation at n^{-1/4} rates), the asymptotic variance of the AIPW estimator is minimized for $μ^1 = μ_1, μ^0 = μ_0$, thus the IPW estimator is generally dominated by the AIPW if $μ^1, μ^0$ are consistent."
> - 4
>
> We have replaced the Angrist and Imbens (1995) reference by "Hernán MA, Robins JM (2020). Causal Inference: What If. Boca Raton: Chapman & Hall/CRC."
>
> We understand that Imbens (2004) may still be appropriate, given that it is mainly about ATEs/ATTs, but we are happy to include a different reference if you believe there are better suited options.
> - 5
>
> Thanks! We have included "In fact, the doubly robust kernel mean embedding estimator may be viewed as an augmented version of the KTE (which is a kernelized IPW) using regression approaches to kernel mean embeddings (Singh et al. 2020), just as AIPW augments IPW with regression approaches" in line 214.
> - 6
>
> We have included "Note that the proposed statistic is, at heart, a two sample test (with a nontrivial causal twist); in contrast to Shekhar et al. (2022), the two samples are not independent and are potentially confounded." in line 222.
>
> We refrained from including Fawkes et al (2022) in the discussion given that we were unable to control the type 1 error of their test (see comment below). They did not present any theoretical guarantee of their current test, and it is still a preprint version -so we understand their method is subject to change-.
> - 7
>
> We agree, sorry about that. We have changed the $\mu$'s in section 3.4 to $\theta$'s, in accordance with the introduction. We have changed the $\mu$_1 and $\mu$_2 (and $\hat \mu_1$ and $\hat \mu_2$) from Appendix C to $\tau_1$ and $\tau_2$  (and $\hat \tau_1$ and $\hat \tau_2$). Consequently, the $\mu$ will solely refer to kernel mean embeddings. We have further changed the generic $\phi$'s in Appendix C for $\omega$'s so that $\phi$ only refers to the AIPW estimator of the mean embedding.
> - 8
>
> With the aforementioned change in lines 190-191, we now point the reader to the appendix where we clarify the notation used.
>
> Further, we have included the sentence "Condition (iv) is equivalent to two-fold cross-fitting i.e. training $\hat \phi ^{(r)}$ on only half of the data and evaluating such an estimator on the remaining half."
> in line 209 before "Condition (v)...".
> - 9
>
> Thanks, we have replaced it by $k(y, \tilde y) = \langle k(\cdot, y), k(\cdot, \tilde y) \rangle$ in line 176 (given that the kernel is going to be evaluated in $\mathcal Y$).
> - 10
>
> Thanks, done.
> - 11
>
> We implemented the test presented in Fawkes (2022), but we obtained rejection rates close to 35% under the null (i.e. we were unable to control the type 1 error at the targeted 5%). They did not prove that their test controls type 1 error, and indeed we found that it does not. Consequently, we left such a test out of the discussion.
> - 12
>
> If the actual embedding was known, the test would be minimax rate optimal against L2 alternatives -inherited from the properties of cross U-statistics, see Proposition E.1 and subsequent comments in Kim and Ramdas (2023)-. Given that we have to estimate the embeddings, the analysis is not straightforward. We refer the reviewer to the paragraph that expands from line 238 to line 245, where we comment on the statistical efficiency.
> - 13
>
> Yes, we have included the 2, thanks.
> - 14
>
> Indeed, the final expression in (13) should have \lambda_1^2. The correction continues throughout the proof without further implications. We have replaced \lambda_1 by \lambda_1^2 in the remaining lines of step 3 and highlighted that \lambda_1 > 0 implies \lambda_1^2 > 0. Thank you for spotting the typo!
> - 15
>
> We have that the square of the sum of non-negative eigenvalues is strictly greater than zero, hence one of them has to be strictly positive, so the first one is strictly positive -by decreasing order of the eigenvalues-. We have included this remark in the proof.

---

### Official Review · Reviewer_gJy9 · 2023-07-06

**Soundness:** 3 good
**Presentation:** 3 good
**Contribution:** 1 poor
**Rating:** 6
**Confidence:** 1

**Summary:**

The paper focuses on studying Augmented Inverse Probability Weighting (IPW) for distributions instead of means. The outline of the paper is as follows:

1. The authors provide motivation for the problem.
1. They review several tools used to solve the problem, including Maximum Mean Discrepancy, Conditional Mean Embeddings, Kernel Treatment Effect, xMMD, and the asymptotics of AIPW.
1. The main results for AIPW in Hilbert spaces are presented, and the authors discuss practical details of the proposed test.
1. Discussion of the experiments.

**Strengths:**

The paper is technical but clearly written. Authors prove non-trivial (to me) technical results, which convinces me that if I were to utilize AIPW for distributions, I would choose this particular implementation. More broadly, if I needed to test treatment effects on distributions and lacked access to propensity scores, I would opt for this test.

**Weaknesses:**

The main limitation of this paper is that I struggle to think of a practical scenario in which I would have an interest in testing differences in the distribution of treatment effects. While the authors briefly mention that this question arises in various applications, none of those applications are utilized in the experiments. I'm uncertain whether investigating the effect of specialist home visits on cognitive test scores, beyond an increase in the mean, is a particularly relevant question to explore.

It might be more appropriate to compare this method with conditional average treatment effect tests. I can imagine situations where there are heterogeneous treatment effects that, on average, cancel each other out but work in opposing directions within two populations.

edit: see the list attached by authors.

**Questions:**

I'd ask authors to discuss applications in which this test would be useful.

**Limitations:**

yes

---

> ### Author Rebuttal · Authors · 2023-08-08
>
> We thank the reviewer for the comments. We would like to address the following weaknesses and questions raised by the reviewer.
>
> - The main limitation of this paper is that I struggle to think of a practical scenario in which I would have an interest in testing differences in the distribution of treatment effects. I'd ask authors to discuss applications in which this test would be useful.
>
> Our work is the extension of kernel two sample tests [1] to the observational causal inference context. In non-causal settings, kernel distributional tests have had a very large impact in the ML community over the last decade (see for instance the citations of [1]). The main advantage of these tests is that they look beyond the mean effect. In causal inference, certainly testing if the average treatment effect equals zero is the most popular way to assess if a treatment is different from a placebo. But one may argue that this is not suitable in many settings, and we should call a treatment as different if (for example) the variance of the response is different from that for the placebo. Kernel treatment effects are a general way to answer these types of questions (with means and variances being special cases obtained with the linear or quadratic kernels).
>
> [1] Gretton, Arthur, et al. "A kernel two-sample test." The Journal of Machine Learning Research 13.1 (2012): 723-773.
>
> - While the authors briefly mention that this question arises in various applications, none of those applications are utilized in the experiments.
>
> We mention that the test may be used to understand whether a treatment simply shifts the distribution of the outcome or, in turn, it also affects higher order moments. This is exactly what we did when investigating the effect of specialist home visits on cognitive test scores.
>
>
> - I'm uncertain whether investigating the effect of specialist home visits on cognitive test scores, beyond an increase in the mean, is a particularly relevant question to explore.
>
> If a treatment simply shifts the distribution of the outcome (in this case, specialist home visits increase cognitive test scores on average) but the higher order moments are unchanged, we can conclude that the treatment is always desired. Otherwise, the treatment may shift the mean but hugely increase the variance, for instance, which may be harmful for the children that are negatively affected by the treatment.
>
>
> - It might be more appropriate to compare this method with conditional average treatment effect tests. I can imagine situations where there are heterogeneous treatment effects that, on average, cancel each other out but work in opposing directions within two populations.
>
> Our test does not attempt to estimate conditional average treatment effects of any form, so we believe that the comparison would not be suitable. Tests for conditional average treatment effects are interesting, but complementary to the current paper.

---

> > ### Comment · Reviewer_gJy9 · 2023-08-17
> >
> > Respectfully, I don't think authors have provided a practical scenario or hypothetical in which I would like to use this test. Perhaps this theoretical construction will find its uses in the future.
> >
> > If the only criterion would be technical novelty I would accept this paper.

---

> > > ### Author Response · Authors · 2023-08-18
> > >
> > > We would like to thank the reviewer again for their time. Although we focused on the specialist home visit example as a novel use of our test, we would like to highlight other potential uses, such that
> > > - Determining subgroups of patients that respond differently to medication and establish treatment policies, referring the reader to [1].
> > > - Conducting feature selection for discovering treatment effect modifiers, referring the reader to [2] and [3].
> > > - Studying the effect of various features on Google advertisers' spending, which has very heavy tails since there are a few advertisers who spend a lot [4]. For this reason means are not very useful summaries and instead distributional effects make a lot more sense.
> > >
> > > We are happy to include any of these references if the reviewer considers that they shed light on the usefulness of our test. Furthermore, we can point the reader to [5], whose introduction discusses some specific motivation for studying distributional effects beyond the mean, as well as the vast literature on quantile treatment effects, which treat distributional effects with the exact same motivation while considering a different distributional target.
> > >
> > > [1] Chikahara, Yoichi, Makoto Yamada, and Hisashi Kashima. "Feature selection for discovering distributional treatment effect modifiers." Uncertainty in Artificial Intelligence. PMLR, 2022.
> > > [2] Bellot, Alexis, and Mihaela van der Schaar. "A kernel two-sample test with selection bias." Uncertainty in Artificial Intelligence. PMLR, 2021.
> > > [3] Biesecker, Leslie G. "Hypothesis-generating research and predictive medicine." Genome research 23.7 (2013): 1051-1053.
> > > [4] Díaz, Iván. "Efficient estimation of quantiles in missing data models." Journal of Statistical Planning and Inference 190 (2017): 39-51.
> > > [5] Kennedy, Edward H., Sivaraman Balakrishnan, and Larry Wasserman. "Semiparametric counterfactual density estimation." arXiv preprint arXiv:2102.12034 (2021).

---

> > > > ### Comment · Reviewer_gJy9 · 2023-08-21
> > > >
> > > > Thank you for putting this list together, I stand corrected.

---

### Official Review · Reviewer_sucV · 2023-07-09

**Soundness:** 2 fair
**Presentation:** 2 fair
**Contribution:** 2 fair
**Rating:** 7
**Confidence:** 4

**Summary:**

The paper introduces a test for the treatment effect which also takes distributional changes into account. The test strongly builds upon the recent works ( Kim and Ramdas, 2023) and ( Muandet et al. (2021)). The main novelty arises from extending the test in Kim and Ramdas, 2023 to the setting of treatment effect estimation. In comparison with previous tests (Figure 4) in the literature, the test proposed in this paper is computationally more efficient at a moderate price in power.

**Strengths:**

Testing for treatment effects is an important problem. The proposed test is computationally efficient and appears to be practically useful.


**Weaknesses:**


1) Figure 2 is misleading to some degree. BART and Causal Forests are estimating the mean of the treatment effect and therefore necessarily fail in scenarios III and IV.

2) I am expecting a comparison against KTE ( Muandet et al. (2021)) and the AIPW extension from  Fawkes et al. (2022)  in the main text. This should come instead of the current  Figure 2. As far as I understand from Figure 4, the main contribution of this paper is not a better test in terms of power but in terms of computational run-time. This does not sufficiently come across in the main text.

3) The contribution of the paper feels fairly limited and more like a straightforward extension of Kim and Ramdas, 2023. The theoretical results appear to me to follow from standard well-established arguments. Can the authors comment more on the technical challenges behind the proofs/ similarities to prior works?

4) While 3) itself is not a reason for a low grade, based on this limitation I would expect a more exhaustive experimental analysis of the method demonstrating the practical usefulness and its limitations on real-world data sets.




**Questions:**

- How does the corresponding plot for Scenario (I) look for the setting in Figure 2 and 4?

- I am willing to increase my score if the authors can address the shortcomings above.

**Limitations:**

See Weaknesses.

---

> ### Author Rebuttal · Authors · 2023-08-08
>
> We thank the reviewer for the comments. We would like to address the following weaknesses and questions raised by the reviewer.
>
> - Figure 2 is misleading to some degree. BART and Causal Forests are estimating the mean of the treatment effect and therefore necessarily fail in scenarios III and IV.
>
> Given that our work is the first to test the KTE in observational settings, there is no algorithm to use for benchmarking, as commented in line 266. We included algorithms designed for the average treatment effect, given that we saw no other choice. And we did say in line 274: "However, and as expected, such methods show no power if the distributions differ but have equal means".
>
>
> - I am expecting a comparison against KTE ( Muandet et al. (2021)) and the AIPW extension from Fawkes et al. (2022) in the main text. This should come instead of the current Figure 2. As far as I understand from Figure 4, the main contribution of this paper is not a better test in terms of power but in terms of computational run-time. This does not sufficiently come across in the main text.
>
> As explained to Reviewer 1 as well, the test presented in (Muandet et al., 2021) cannot be used with unknown propensity scores, and hence it does not apply to our problem (observational studies). We only included a comparison between KTE and AIPW-xKTE in Appendix A to show that, in the specific case of having known propensity scores (experimental studies), our test loses a little power for a large computational gain.
>
> Further, we implemented the test presented in Fawkes (2022) -still a preprint-, but we obtained rejection rates close to 35% under the null (i.e. we were unable to control the type 1 error). This is not surprising: if you read carefully, Fawkes (2022) does not present any form of theoretical guarantee of the test. They do not claim that their test controls type-1 error, and indeed we find that it does not. Consequently, we left such a test out of the discussion in order to not be overly critical about a preprint, but we can add a note about our experiments with it.
>
> Our test is the only test with theoretical type 1 error control in observational studies (where propensity scores are not known).
>
>
> - The contribution of the paper feels fairly limited and more like a straightforward extension of Kim and Ramdas, 2023. The theoretical results appear to me to follow from standard well-established arguments. Can the authors comment more on the technical challenges behind the proofs/ similarities to prior works?
>
> Although we presented the contribution as if it was a natural extension of previous works, we would like to highlight the nontrivial technical challenges addressed in Appendix C. The proof of Theorem 4.1, which is around 6 pages long, is completely novel, and cannot reduce to invoking any existing theorems. It combines the main idea presented in Kim and Ramdas, 2023 with a variety of techniques including kernel ideas, functional data results and causal inference results (eg: the novel Lemma C.7 and Theorem C.9) .
>
> We highlight that the proof heavily differs from the work presented in Kim and Ramdas (2023) and any other paper that we have read on the topic. In fact, the only relevant part from their paper is Equation 14 and Equation 15 (part of step 4). We thus emphasize that the theoretical contribution of the work is far from being straightforward. We have now added a paragraph in the main paper to highlight these advances.
>
>
>
> - While 3) itself is not a reason for a low grade, based on this limitation I would expect a more exhaustive experimental analysis of the method demonstrating the practical usefulness and its limitations on real-world data sets.
>
> We considered, in total, 8 different scenarios with synthetic data and 6 different scenarios with real data. Further, we highlight that it is nearly impossible to assess our test in real-life observational studies, given that we do not know the ground truth. The test may reject or accept the null hypothesis, but we have no information on whether the null hypothesis is actually true or not. These experiments vary different aspects of the considered method and competitors. But if there is a particular aspect that you would like to see better explored, we would be happy to add an experiment for it.
>
>
> - How does the corresponding plot for Scenario (I) look for the setting in Figure 2 and 4?
>
> The Gaussian behavior (Subfigure A and Subfigure B)  is only expected with our proposed test, so those subfigures would not be interesting for the remaining tests. Subfigure C is very similar for all the tests studied (around 0.05 with some noise), which is expected given that they are all provably well calibrated (hence we refrained from including those in the paper). We can add a note about these to the paper.

---

> > ### Comment · Reviewer_sucV · 2023-08-10
> > **Response to the rebuttal**
> >
> > I would like to thank the authors for their response. I have increased my score to a 7.
> >
> > Furthermore, I agree with the following concern raised by a different reviewer:
> >
> > >The main limitation of this paper is that I struggle to think of a practical scenario in which I would have an interest in testing differences in the distribution of treatment effects. While the authors briefly mention that this question arises in various applications, none of those applications are utilized in the experiments. I'm uncertain whether investigating the effect of specialist home visits on cognitive test scores, beyond an increase in the mean, is a particularly relevant question to explore.
> >
> > It would be beneficial to provide a concrete application as an example to highlight the necessity of this test. However, considering the limited access to public datasets, this might be an ambitious request.

---

> > > ### Author Response · Authors · 2023-08-18
> > >
> > > Following the concerns raised by Reviewer gJy9, we have provided further examples of use of our test in the respective official comment. We would like to thank the reviewer again for their insightful comments.

---

### Official Review · Reviewer_yX9h · 2023-07-14

**Soundness:** 4 excellent
**Presentation:** 2 fair
**Contribution:** 2 fair
**Rating:** 4
**Confidence:** 4

**Summary:**

The paper introduces a statistical test to determine whether the distributions of the two counterfactuals are the same. This goes beyond the well-known average treatment effect, which only tries to understand whether the means of the distributions are the same. The first work in this direction was by (Muandet et al., 2021), who introduced the Kernel Treatment Effect (KTE). However, their test statistic is degenerate, which means that they cannot use the CLT to derive an asymptotic threshold, and they need to resort to a permutations approach in order to compute the threshold. This paper introduces the AIPW-xKTE test, which generalizes KTE by including a plug-in estimator in analogy with AIPW vs IPW, and most importantly by using the same approach as the Cross MMD test from (Kim and Ramdas, 2023), which does yield a statistic with asymptotically normal distribution.


**Strengths:**

The contribution of the paper is clear, and the authors do a reasonable job at placing it within the literature.


**Weaknesses:**

The main weakness is that the contribution is not highly novel, in that the test proposed is basically a combination of the KTE test from (Muandet et al., 2021) and the Cross MMD technique from (Kim and Ramdas, 2023).

The explanation would be more transparent if some important concepts were clearly defined. See more details in the questions section. There are some more detailed weaknesses that I also point out in the questions section.


**Questions:**

- What is double robustness? It is a relevant concept in the paper, as it is mentioned twice in the contributions part of the introduction, and many more times later on. However, it is never defined.

- Line 23: The plug in estimator is not well defined: what does it actually look like? In lines 222-225, the estimator appears again under a different notation (\hat{\beta} instead of \hat{\theta}). The authors currently say that “At this time, not so many choices exist for estimators…“ and they cite a work on this. It would be good to give a more detailed explanation of how the estimator \hat{\beta} is computed. Similarly, it would be helpful to give more insight on how \hat{\pi} is computed.

- Theorem 3.1: What is \hat{\pi}? What is \hat{\psi}_{DR}? These quantities have not been defined before as far as I can tell.

- Figure 1: Show error bars in subfigure c, to show that the discrepancy from 0.05 can be attributed to a statistical error. The key question that needs to be answered here is: how large does n need to be for the CLT to kick in and for the Type I error guarantee to hold. Without error bars, the current figure does not provide an answer to this question.

- Figure 2: Although not as critical, error bars in this figure would be appreciated too.

- Table 1: Show standard error.

- Comparison with (Muandet et al., 2021): The authors do not show any experimental comparison with the KTE test proposed by (Muandet et al., 2021). They argue that “Due to the fact that the KTE (Muandet et al., 2021) may not be used in the observational setting, where the propensity scores are not known, there is no natural benchmark for the proposed test.” Since AIPW-xKTE makes use of estimators of propensity scores, it seems natural to me to compare it with KTE using the same propensity score estimators. Is there a reason not to do this?


**Limitations:**

No limitations

---

> ### Author Rebuttal · Authors · 2023-08-08
>
> We thank the reviewer for the comments. We would like to address the following weaknesses and questions raised by the reviewer.
>
> - The main weakness is that the contribution is not highly novel, in that the test proposed is basically a combination of the KTE test from (Muandet et al., 2021) and the Cross MMD technique from (Kim and Ramdas, 2023).
>
> Although we presented the contribution as if it was a natural extension of previous works, we would like to highlight the nontrivial technical challenges addressed in Appendix C. The proof of Theorem 4.1, which is around 6 pages long, is completely novel, and cannot reduce to invoking any existing theorems. It combines the main idea presented in Kim and Ramdas, 2023 with a variety of techniques including kernel ideas, functional data results and causal inference results (eg: the novel Lemma C.7 and Theorem C.9) .
>
> We highlight that the proof heavily differs from the work presented in Kim and Ramdas (2023) and any other paper that we have read on the topic. In fact, the only relevant part from their paper is Equation 14 and Equation 15 (part of step 4). We thus emphasize that the theoretical contribution of the work is far from being straightforward. We have now added a paragraph in the main paper to highlight these advances.
>
> - What is double robustness? It is a relevant concept in the paper, as it is mentioned twice in the contributions part of the introduction, and many more times later on. However, it is never defined.
>
> We have changed line 33 from
> "... we highlight that double-robustness is a property that is shared with many other estimators."
> to
> "... we highlight that double-robustness is an intriguing property of an estimator that makes use of two models, in which the estimator is consistent even if only one of the two models is well-specified and the other may be misspecified; we refer the reader to [1] for a discussion on doubly-robust procedures."
> [1] Kang, Joseph DY, and Joseph L. Schafer. "Demystifying double robustness: A comparison of alternative strategies for estimating a population mean from incomplete data." (2007): 523-539.
>
> - Line 23: The plug in estimator is not well defined: what does it actually look like? In lines 222-225, the estimator appears again under a different notation ($\hat\beta$ instead of $\hat\theta$). The authors currently say that “At this time, not so many choices exist for estimators…“ and they cite a work on this. It would be good to give a more detailed explanation of how the estimator $\hat\beta$ is computed. Similarly, it would be helpful to give more insight on how $\hat\pi$ is computed.
>
> The plug in estimator is well defined (line 24), but its form depends on the choice of the regression functions (denoted $\hat\theta$). Throughout the work, we denote univariate or multivariate regressors by $\hat\theta$, and infinite-dimensional regressors (such as conditional mean embeddings) by $\hat\beta$, which is introduced in line 183.  Thank you for pointing out that we have not explicitly mentioned this somewhere, we now do so in Section 4 after line 179.
>
> Many different options exist for $\hat\theta$, $\hat\beta$, and $\hat\pi$. Linear regression, logistic regression, random forests, or neural nets may be used for $\hat{\theta}$ and $\hat{\pi}$; conditional mean embeddings or the work cited in line 225 may be used for $\hat\beta$.
>
> - Theorem 3.1: What is $\hat\pi$? What is $\hat\psi_{DR}$?
>
> $\hat{\pi}$ is defined in line 27. $\hat\psi_{DR}$ should be $\hat\psi_{AIPW}$  instead, defined in line 28 (we thank the reviewer for spotting this typo!).
>
> - Figure 1: Show error bars in subfigure c, to show that the discrepancy from 0.05 can be attributed to a statistical error. The key question that needs to be answered here is: how large does n need to be for the CLT to kick in and for the Type I error guarantee to hold. Without error bars, the current figure does not provide an answer to this question
> Figure 2: Error bars in this figure would be appreciated too
> Table 1: Show standard error.
>
> The outcome of the test is either a 0 or a 1, hence its distribution is fully specified by its mean, which is estimated by Monte Carlo (repeating the test B times for a large B). The only error is Monte Carlo error, and we have now added error bars to show that our observations are not a result of Monte Carlo error. We uploaded a PDF with some plots in the Author Rebuttal section.
>
> - Comparison with (Muandet et al., 2021): The authors do not show any experimental comparison with the KTE test proposed by (Muandet et al., 2021). They argue that “Due to the fact that the KTE (Muandet et al., 2021) may not be used in the observational setting, where the propensity scores are not known, there is no natural benchmark for the proposed test.” Since AIPW-xKTE makes use of estimators of propensity scores, it seems natural to me to compare it with KTE using the same propensity score estimators. Is there a reason not to do this?
>
> The test proposed in (Muandet et al., 2021) is not valid if the propensity scores are misspecified (which is usually the case when the propensity scores are unknown). In causal inference, there is a huge difference between randomized experiments (where they are known) and observational settings (where they are unknown). Plugging in estimated propensity scores for the true ones doesn’t work: that would be too easy a solution to the latter problem — this is why the AIPW estimator is not simply the IPW estimator with plug-in estimate of the propensity score: one has to “debias” IPW in some sense and also reduce its variance. Muandet’s test only works in the randomized experiments setting, while ours is designed for the much harder observational setting. Consequently, we refrained from including it in the main body of the paper, where we discuss observational studies. We included a comparison between KTE and AIPW-xKTE in Appendix A, for the specific case of having known propensity scores (experimental studies).

---

### Author Rebuttal · Authors · 2023-08-08

We upload a PDF containing the changes in Figure 1, Figure 2 and Table 1, now with error bars / standard errors, suggested by one of the reviewers. We have included error bars in the remaining figures of the paper, not shown in here for space constraints.

---

### Decision · Program_Chairs · 2023-09-21

**Decision:**

Accept (poster)

**Comment:**

The paper makes timely contribution by expanding the field of MMD based testing for usage of testing in causal inference problems and provides an elegant doubly robust solution without needing permutations. The reviewers agreed and had a consensus towards accept.